# First Evidence of the Post-Variscan Magmatic Pulse on the Western Edge of East European Craton: U-Pb Geochronology and Geochemistry of the Dolerite in the Lublin Podlasie Basin, Eastern Poland

Ewa Krzemińska [1,*], Leszek Krzemiński [1], Paweł Poprawa [2], Jolanta Pacześna [1] and Krzysztof Nejbert [3]

[1] Polish Geological Institute—National Research Institute, 00-975 Warsaw, Poland;
leszek.krzeminski@pgi.gov.pl (L.K.); jolanta.paczesna@pgi.gov.pl (J.P.)
[2] Faculty of Geology, Geophysics and Environmental Protection, AGH University of Science and Technology, 30-059 Cracow, Poland; ppop.ecr@gmail.com
[3] Department of Geology, University of Warsaw, 02-089 Warsaw, Poland; knejbert@uw.edu.pl
[*] Correspondence: ewa.krzeminska@pgi.gov.pl

**Abstract:** The U–Pb measurements of youngest, coherent group of zircons from the Mielnik IG1 dolerite at the Teisseyre-Tornquist margin (TTZ) of East European Craton (EEC) in Poland yielded age of $300 \pm 4$ Ma. Zircon dated an evolved portion of magma at the late stage crystallization. It is shown that this isolated dyke from the northern margin of the Lublin Podlasie basin (Podlasie Depression) and regional dyke swarms of close ages from the Swedish Scania, Bornholm and Rügen islands, Oslo rift, Norway, and the Great Whine Sill in northeastern England, were coeval. They have been controlled by the same prominent tectonic event. The Mielnik IG1 dolerite is mafic rock with Mg-number between 52 and 50 composed of the clinopyroxene, olivine-pseudomorph, plagioclase, titanite, magnetite mineral assemblage, indicating relatively evolved melt. This hypabyssal rock has been affected by postmagmatic alteration. The subalkaline basalt composition, enrichment in incompatible trace elements, progressive crustal contamination, including abundance of zircon xenocrysts determines individual characteristics of the Mielnik IG1 dolerite. The revised age of dolerite, emplaced in vicinity of TTZ provides more evidences documenting the reach of the Permo-Carboniferous extension and rifting accompanied by magmatic pulses, that were widespread across Europe including the margin of the EEC incorporated that time into the broad foreland of the Variscan orogen.

**Keywords:** sub-alkaline basalt; zircon; late Carboniferous; Variscan foreland; extension; Teisseyre-Tornquist margin

## 1. Introduction

The south-western part of East European Craton (EEC), after the collision of three proto-cratons: Volgo-Uralia, Sarmatia and Fennoscandia, had been completed at ca. 1.82–1.80 Ga [1] and has become a relatively stable continental region. However its cratonised structure has been complicated by faults, particularly rift zones. Their distribution provides information about the spatial and temporal changes of ancient crustal structures, the regional stress field, their intensity and reasons for them. They became pathways for the injections of melts, forming a single dyke or swarms of dykes.

The crystalline basement of Fennoscandia and its sedimentary cover was pierced by a few generations of dykes. At ca 1.27 Ga mafic sills and dykes in the central part of Scandinavia, represented by the Västerbotten, Ulvö, Jämtland, Satakunta and Dalarna "post-Jotnian" dolerite, intruded in response to the rifting of Laurentia/Baltica [2,3].

Afterwards, the Late Ediacaran extensional tectonic regime along the SW margin of Baltica resulted in a network of rift basins during the fragmentation of Rodinia. The

rift-related sedimentary and volcanic sequence resulted in mafic dyke swarms on the Baltoscandian margin [4] and, somewhat later, in flood basalts and dolerites of westernmost Ukraine, Belorussia and eastern Poland [5–7]. The subsequent opening of the Tornquist Ocean and the development of passive margin-type sedimentary basins with Ediacaran-early Palaeozoic, Devonian-Carboniferous, and Permian-Mesozoic sediments, were punctured by a few successive extension phases accompanied by short igneous episodes (Figure 1).

The existence of thousands of WNW- to NW-trending dykes, of various thicknesses and lengths, is known (Figure 1) in the southernmost Swedish province of Scania [8–10] Västergötland in central Sweden and the Danish island of Bornholm [11]. They intrude the granites and gneisses of the Precambrian basement and the overlying Lower Paleozoic sediments. Further south-east in the area of North-Eastern Poland the dykes are not so common.

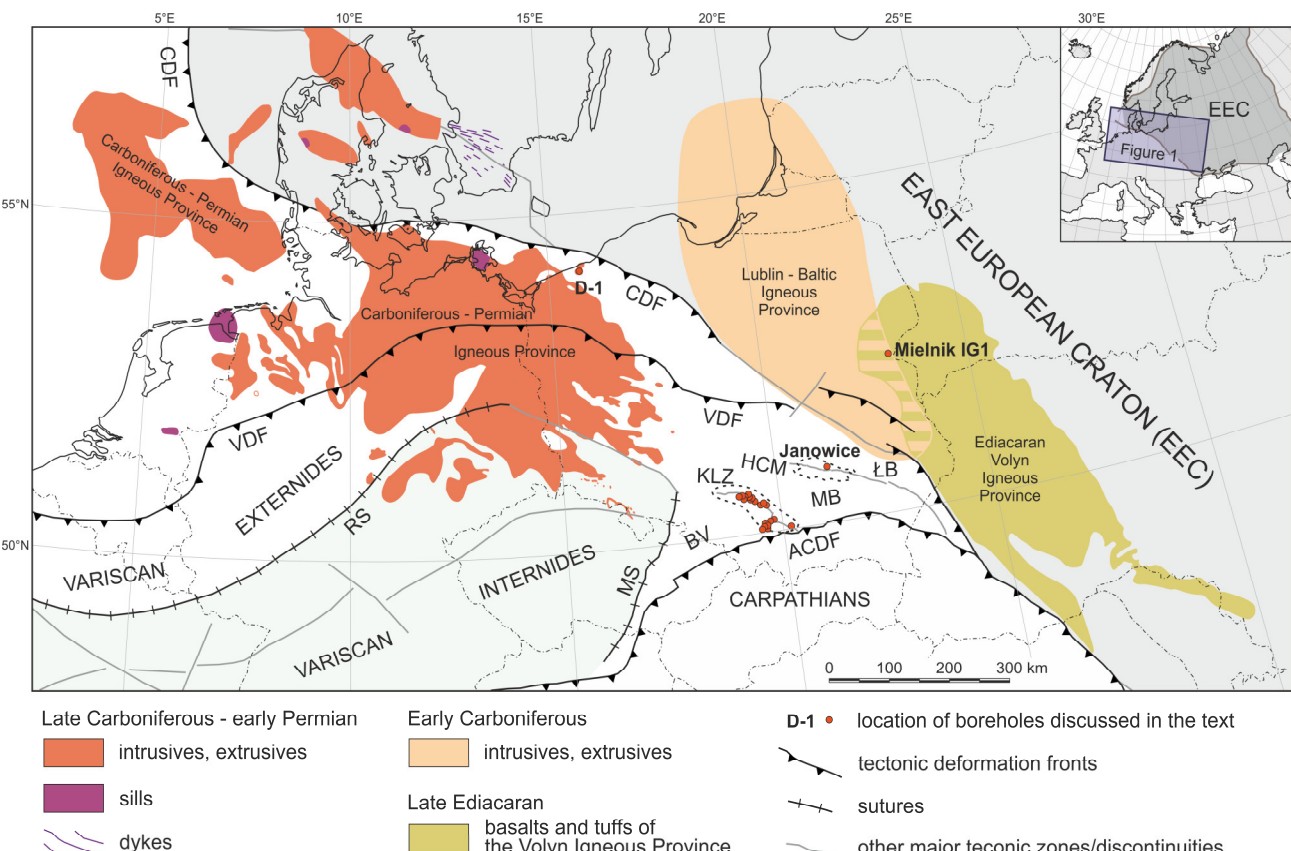

**Figure 1.** Location of the study area against the major igneous provinces of the central and western Europe. The Carboniferous-Permian igneous province of the central and north-western Europe after [12,13]. The alkaline Lublin-Podlasie Mississippian Igneous province after [14], The Volyn Igneous Province after [14,15]. Tectonic framework—compilation after [14,16–18] Mielnik IG1, J-2—Janowice 2, HCM—Holy Cross Mountains, KLZ—Kraków-Lubliniec Zone, MS—Moravian Suture, RS—Rheic Suture, VDF—Variscan Deformation Front, CDF—Caledonian Deformation Front, ACDF—Alpine-Carpathian Deformation Front, ŁB—Łysogóry Block, MB—Małopolska Block, BV—Brunovistulicum.

This contribution presents the only obtained U-Pb zircon ages and whole-rock geochemical characteristics for the dolerite previously known from the Mielnik IG1 drill hole, in North-Eastern Poland in the vicinity of the Teisseyre-Tornquist margin. Until the U-Pb SHRIMP zircon measurements, the ~18 m thick dolerite dyke in the Proterozoic part of the core section was considered to be an example of the Neoproterozoic sub-volcanic rock, which predated Volynian volcanic activity [19,20]. The new U-Pb age investigation significantly changes its stratigraphic position. The revised Late Carboniferous/Early Permian

age requires a new correlations for the Mielnik IG1 dolerite sill. On a regional scale, in the vicinity of Teisseyre-Tornquist margin, the comparison of the sill with well-known dyke swarms in Scania reveals the reach of the post-Variscan extension, illustrating the diversity of dolerite magmatism on the SW cratonic margin of Fennoscandia.

## 2. The Evidence of Dolerites on the SW Slope of EEC

The SW margin of the EEC was disconformably overlain by a Neoproterozoic–lower Paleozoic succession. Since the Mesoproterozoic time, the western segment of the EEC hosted several episodes of continental, rift-related magmatism, resulting in the emplacement of dolerite dykes. The prominent evidence of this type of magmatism within the Paleoproterozoic interior of Fennoscandia that remains is the Central Scandinavian Dolerite group. These mafic dykes intruded the older basement rock between 1271–1246 Ma in response to the distal events of Mesoproterozoic subduction along the westernmost margin of Fennoscandia [21], during the Grenvillian–Sveconorwegian orogeny (1.25–0.9 Ga). The volumetrically-minor dike swarm in Egersund (SW Norway), the Rogaland Anorthosite Complex intruding at $616 \pm 3$ Ma [4] into the surrounding Sveconorwegian terrane as well as the Ottfjället dyke swarm intruding at $596.3 \pm 1.5$ Ma in the sedimentary succession of Scandinavian Caledonides, are interpreted to be related to rifting along the western Baltica margin prior to opening of the Iapetus Ocean [22,23].

In western Belarus and Ukraine and eastern Poland, the effects of the same extension processes and continental rifting were attributed to the diachronic volcanic event forming the Volynian Igneous Province between 570 and 546 Ma [6,7,24]. Dolerites located immediately beneath the Volyn flood basalt sequence, were identified in Belarus and Ukraine [6,25,26], have limited radiometric dating. An olivine dolerite yields a U-Pb baddeleyite age of $626 \pm 17$ Ma by TIMS technique [6], which is an age that predates the main volcanic phase of the Volyn Flood Basalt Province.

In Eastern Poland, the occurrence of dolerite in relation with the flood basalts was described in the Mielnik IG1 drill section. According to the initial reconnaissance [27] the dolerite was assigned a "Jotnian" age, intruding at ca. <1300 Ma. In subsequent publications the Mielnik IG1 dyke was interpreted as genetically related to the volcanic activity of the Late Neoproterozoic [7,19,20].

The magma injections piercing the Paleozoic sedimentary cover are not commonly evidenced in this part of the EEC margin, but a few isolated dolerites of limited thickness were also identified [28]. Some of them forming a dyke swarms are located near the TTZ in NE Poland (Figure 2), in the vicinity of the prominent Carboniferous intra-cratonic alkaline and ultramafic-alkaline massifs of e.g., Ełk, Pisz, Tajno, [29–31]. They appear as several meter thick veins between the Silurian and Rotliegend sediments. The dolerite vein emplacement ages and ones of other subvolcanic rocks (Table 1) from Ciechanów 1, Konopki 1 Ełk IG1, Tajno IG1, Prostki 1, Klusy 1, Bargłów 1, Olsztyn 1, 2, were sometimes determined with poor precision [29]. Most suggest the longer magmatic activity postdating an early Carboniferous emplacement of the Ełk, Pisz and Tajno massifs.

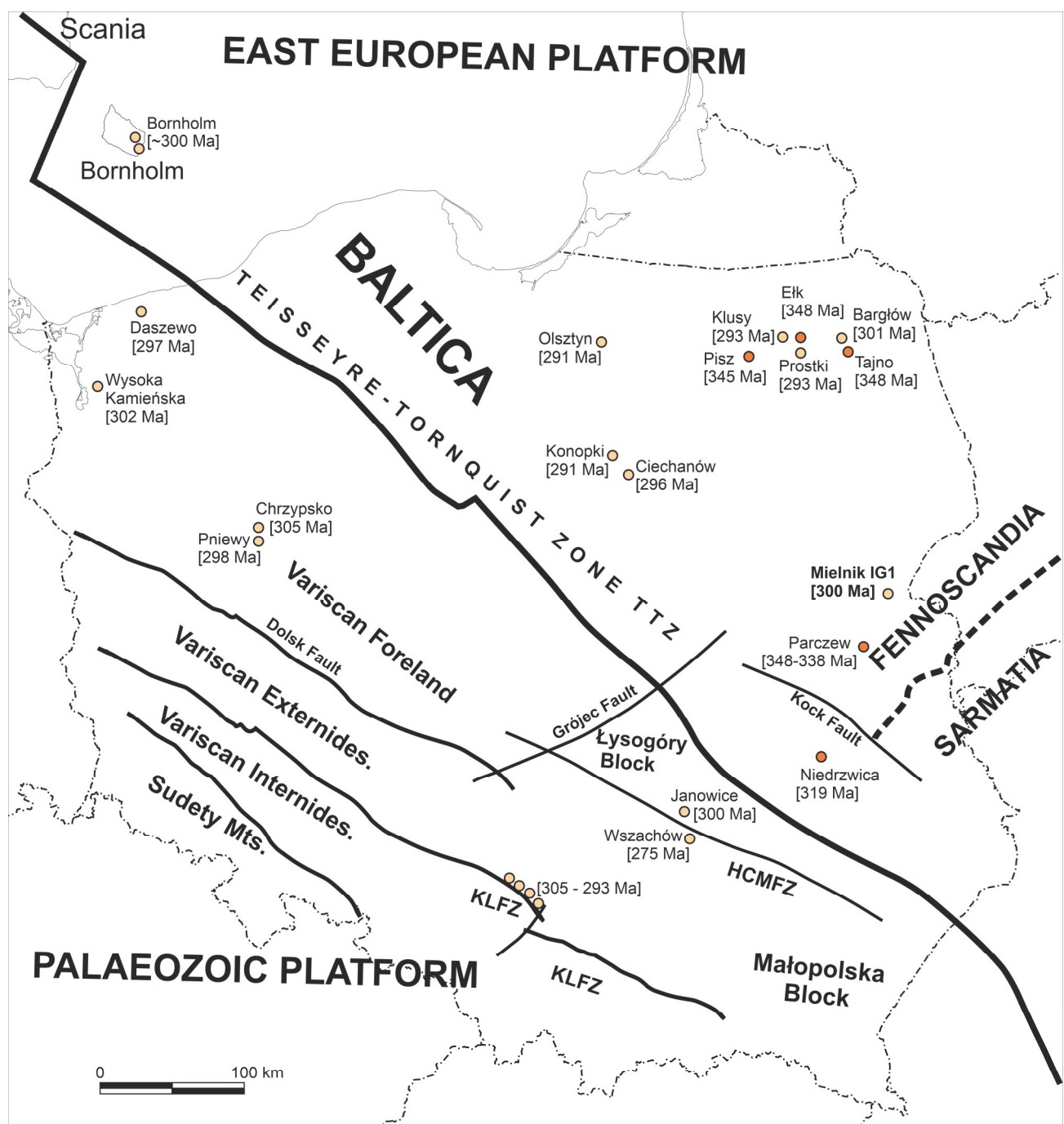

**Figure 2.** The sketch map the SW edge of Baltica showing Mielnik IG1 drill hole location, and regional distribution of early and late Carboniferous intrusions and their age. Data sources: [29–34]; HCMFZ—Holy Cross Mountains Fault Zone, KLFZ—Kraków-Lubliniec Fault Zone.

Further south isolated volcanic rocks (alkaline basalts) were identified in the lowermost part of Carboniferous succession. They are described as dolerites (diabases) or tuffo-lavas, at the Parczew IG7, Parczew IG9, Parczew IG10, Lublin 1, Niedrzwica IG1 and Kolechowice 24 drill sites [32,35]). The whole-rock $^{40}Ar/^{39}Ar$ data for two of them (Parczew IG7 and Parczew IG9) indicate a Carboniferous (Tournaisian) emplacement ages of 348 ± 0.8 Ma and 338.5 ± 0.7 Ma, respectively [32].

**Table 1.** The examples of age data related to Carboniferous–early Permian extensional magmatism—selected dyke swarms and dolerite from northern Europe.

| Sample ID | Coordinates | Type of Rock | Age [Ma] | Method | Ref. |
|---|---|---|---|---|---|
| Great Whin Sill (NE England) | 54°39′ N, 2°11′ W | Dolerite | 297 ± 0.4 | U-Pb | [36] |
| **Fennoscandia** | | | | | |
| Hardeberga (Sk) | 55°42′ N, 13°17′ E | Dolerite | 285 ± 4 | K-Ar | [37] |
| Åstorp (Sk) | 56°08′ N, 12°57′ E | Dolerite | 296 ± 4 | K-Ar | [37] |
| Rönnarp (Sk) | 55°57′ N, 12°57′ E | Dolerite | 298 ± 4 | K-Ar | [37] |
| Mölle (Sk) | 56°17′ N, 12°29′ E | Dolerite | 313 ± 4 | K-Ar | [37] |
| ÖnnEstad (Sk) | 56°02′ N, 14°00′ E | Dolerite | 318 ± 5 | K-Ar | [37] |
| Mösseberg | 58°12′ N, 13°30′ E | Dolerite | 298 ± 2.8 | Ar-Ar | [11] |
| Kinnekulle | 58°36′ N, 13°24′ E | Dolerite | 293 ± 1.3 | Ar-Ar | [11] |
| Billingen | 58°24′ N, 13°46′ E | Dolerite | 300 ± 1.6 297.5 ± 3 | Ar-Ar Ar-Ar | [11] |
| Rügen | 54°21′ N, 13°35′ E | Dolerite | 306 ± 11 | Ar-Ar | [11] |
| Bornholm | | Dolerite | ~300 | K-Ar | [38,39] |
| Olsztyn 1 | 53°47′ N, 20°00′ E | Dolerite | 289 | K-Ar | [29] |
| Olsztyn 2 | 53°53′ N, 19°57′ E | Dolerite | 291 | K-Ar | [29] |
| Bargłów IG1 | 53°45′ N, 22°50′ E | Camptonite | 278 ± 11 301 ± 11 | K-Ar | [30] |
| Klusy 1 | 53°48′ N, 22°10′ E | Syenite | 293 | K-Ar | [29] |
| Prostki 1 | 53°42′ N, 22°22′ E | Syenite | 293 | K-Ar | [29] |
| TajNo IG1 | 53°42′ N, 22°51′ E | Microsyenite | 289 | K-Ar | [29] |
| Ełk IG1 | 53°52′ N, 22°23′ E | Porphyrite | 285 | K-Ar | [29] |
| Ciechanów 1 | 52°49′ N, 20°32′ E | Porphyrite | 296 287 | K-Ar | [29] |
| Konopki 1 | 52°58′ N, 20°24′ E | Porphyrite | 304 291 | K-Ar | [29] |
| Mielnik IG1 | 52°20′ N, 23°01′ E | Dolerite | 300 ± 4 | U-Pb | This study |
| Niedrzwica | 51°06′ N, 22°21′ E | Dolerite | 319 | K-Ar | [29] |
| Lublin 1 | 51°11′ N, 22°41′ E | Dolerite | 333 | K-Ar | [29] |
| Okuniew IG1 | 52°16′ N, 21°17′ E | Dolerite | 356 | K-Ar | [29] |
| **Trans-European Suture Zone** | | | | | |
| Daszewo | 54°03′ N, 15°55′ E | Rhyolite | 297 ± 1 | U-Pb | [34] |
| Wysoka Kamieńska | 53°47′ N, 14°53′ E | Rhyolite | 302 ± 1.5 | U-Pb | [34] |
| Pniewy | 52°33′ N, 16°21′ E | Rhyolite | 298 ± 1.7 | U-Pb | [40] |
| Chrzypsko | 52°39′ N, 16°13′ E | Rhyolite | 302 ± 1 | U-Pb | [34] |
| Janowice 2 | 50°50′ N, 21°13′ E | Dolerite | 300 ± 10 | U-Pb | [41] |
| Milejowice 1 | 50°50′ N, 21°14′ E | Dolerite | 322 ± 1 331 ± 2 | Ar-Ar | [42] |

**Table 1.** *Cont.*

| Sample ID | Coordinates | Type of Rock | Age [Ma] | Method | Ref. |
|---|---|---|---|---|---|
| | | **Małopolska Block** | | | |
| Podkranów | 50°47′ N, 20°46′ E | Lamprophyre | 322 ± 10 | U-Pb | [41] |
| Wszachów | 50°46′ N, 21°09′ E | Lamprophyre | 275 ± 15 | K-Ar | [43] |
| KLFZ [range] | | Volcanic & plutonic | from 293 ± 5 to 305 ± 2 | U-Pb | [44–46] |

KLFZ—Kraków–Lubliniec Fault Zone; Sk—Scania.

## 3. Geological Context of the Mielnik IG1 Dolerite Occurrence

The deep drill hole Mielnik IG1 in which the studied dolerite was identified (Figure 3), belongs to the earliest exploration drilling program implemented by the Polish Geological Institute. It was completed in 1960. The drill core is available in the core repository of the National Geological Archive (Figure 4).

Mielnik IG1 is located in the northern part of the structure known as Lublin-Podlasie Basin (LPB) that belongs to the NW–SE elongated system of Upper Proterozoic-Lower Paleozoic basins, formed on the western slope of the EEC.

The LPB had developed in Late Neoproterozoic as an active rift, but has been aborted and gradually turned into the Cambrian-Ordovician post-rift thermal sag on Baltica's passive margin. Its distinguishing feature is the almost continuous, vertical profile for the Upper Neoproterozoic to Silurian sediments [47,48]. In this part of the EEC the dominant evidence of igneous activity remains, the volcanogenic Sławatycze Formation, in E Poland, being analogous to the Volyn Series, in Ukraine [6,7].

The initial rift-related sediments (Figure 3) are alluvial basal conglomerates and coarse-grained sandstones, referred to as the Żuków Formation [49,50], passing up-section into volcanogenic rocks of the Sławatycze Formation, with the record of explosive and effusive basaltic volcanic activity of vent complexes located on tectonic fissures in rift grabens or half-grabens. There is a large volume of basaltic lava flows 10 to 30 m thick. Both basaltic bodies are overlain by beds of tuff and tuff agglomerate (Figure 3). No erosional surfaces are observed at these contacts. In the upper part of the succession, a large interval of melaphyre and auto-brecciated melaphyre lavas occur. Towards the top of the succession, effusive eruptions are less common. The seven identified explosive episodes represent a large volume of proximal pyroclastic material. The thickness of fine-grained tuff, lapilli tuff and tuff agglomerate beds is variable, ranging from a few meters at the bottom of the succession to 12 m at its top.

The Volyn Series passes disconformably up-section into the uppermost Ediacaran siliciclastic complex, represented by riverine-estuarine sediments deposited in a transgressive cycle, divided into the Siemiatycze, Białopole, Łopiennik and Włodawa Formations [48–50]. In some of these clastic rock layers (Figure 3), the age of the detrital zircon populations has been identified [45]. The thickness of this complex locally exceeds 600 m. It gradually passes up into the Cambrian marine clastic sequence [51]. This sequence terminates with the upper Cambrian hiatus.

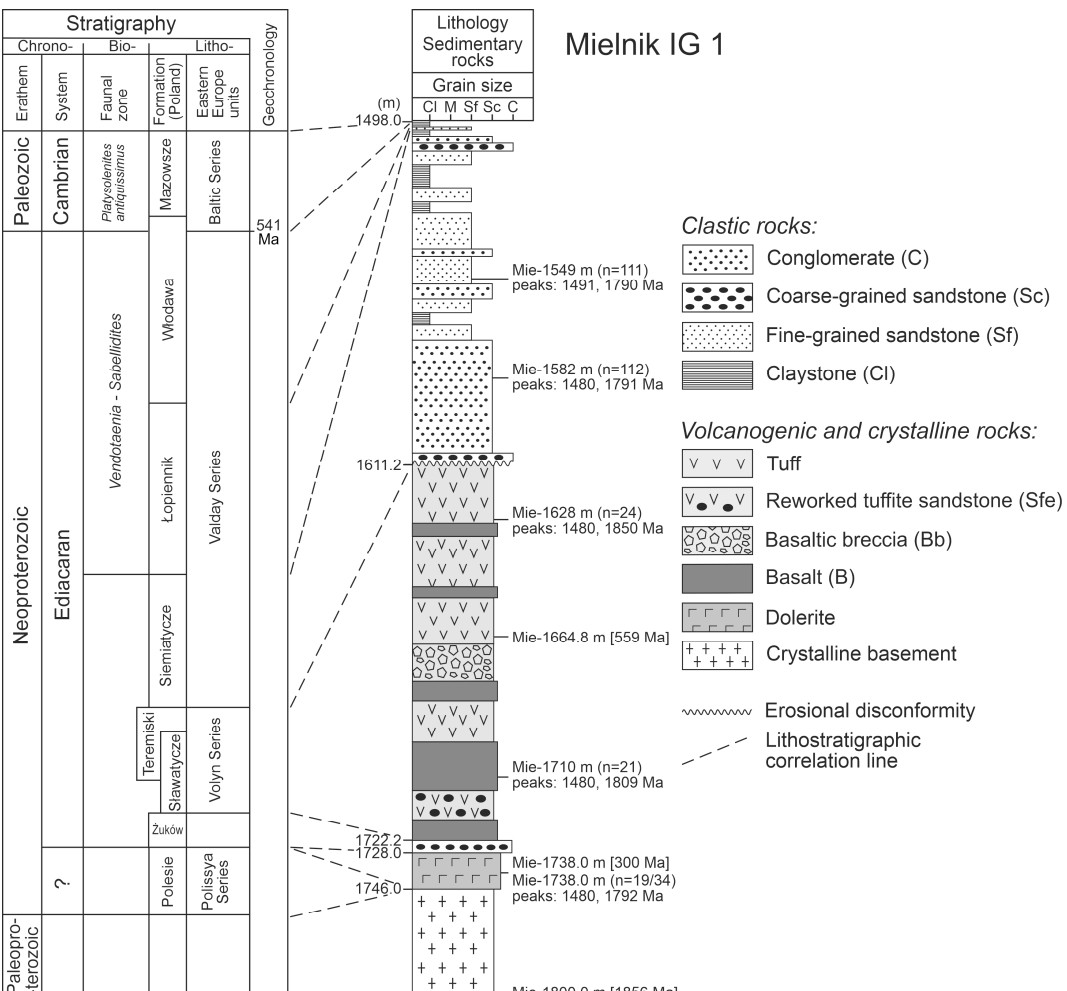

**Figure 3.** Lithostratigraphic section for the bottom part of the Mielnik IG1 drill hole showing location of dolerite sill (depth interval of 1746–1728 m) sampled for age investigations and uppermost Ediacaran volcanogenic and clastic deposits and lowermost Cambrian clastic deposits. The location of previously dated samples including those on clastic rocks [7,33,45] are also indicated. Polish chrono- and lithostratigraphy after [49,50] Ukrainian and Belarusian lithostratigraphy after [15].

During the Ordovician, deposition was continued in form of relatively thin shallow marine clastic and carbonate sediments overlain by the deposition of Silurian marine shales and marly shales, characterized by high thickness, locally exceeding 1300 m [52]. In the NW of the Lublin Basin, over a vast part of the EEC, the Devonian and Carboniferous sediments were removed by erosion. A regional unconformity separates the Devonian–Carboniferous sedimentary complex from the Permian–Mesozoic complex across all the area considered. In the Mielnik IG1 section, a hypabyssal mafic rock lies between the top of the Paleoproterozoic (1.86 Ga) crystalline basement rocks, at a depth of 1746 m, but before the clearly identified Żuków Formation at a depth 1728 m. The dolerite lies above as well as at the bottom of Volyn volcanic series, at the depth of 1722 m (Figure 3). A crucial contact between the Mielnik IG1 dolerite and the Żuków Formation conglomerate has not been properly identified and maintained (Figure 4A), leading to confusion in the literature about its likely age [19,20,49,50].

The dolerite has a greenish rusty brown colour (Figure 4B). It shows a monotonous ophitic texture, with randomly oriented plagioclase laths enclosed by clinopyroxene (augite) and Fe oxides. An archival description [27] indicates finer-grained forms at the bottom of sill (chilled margin).

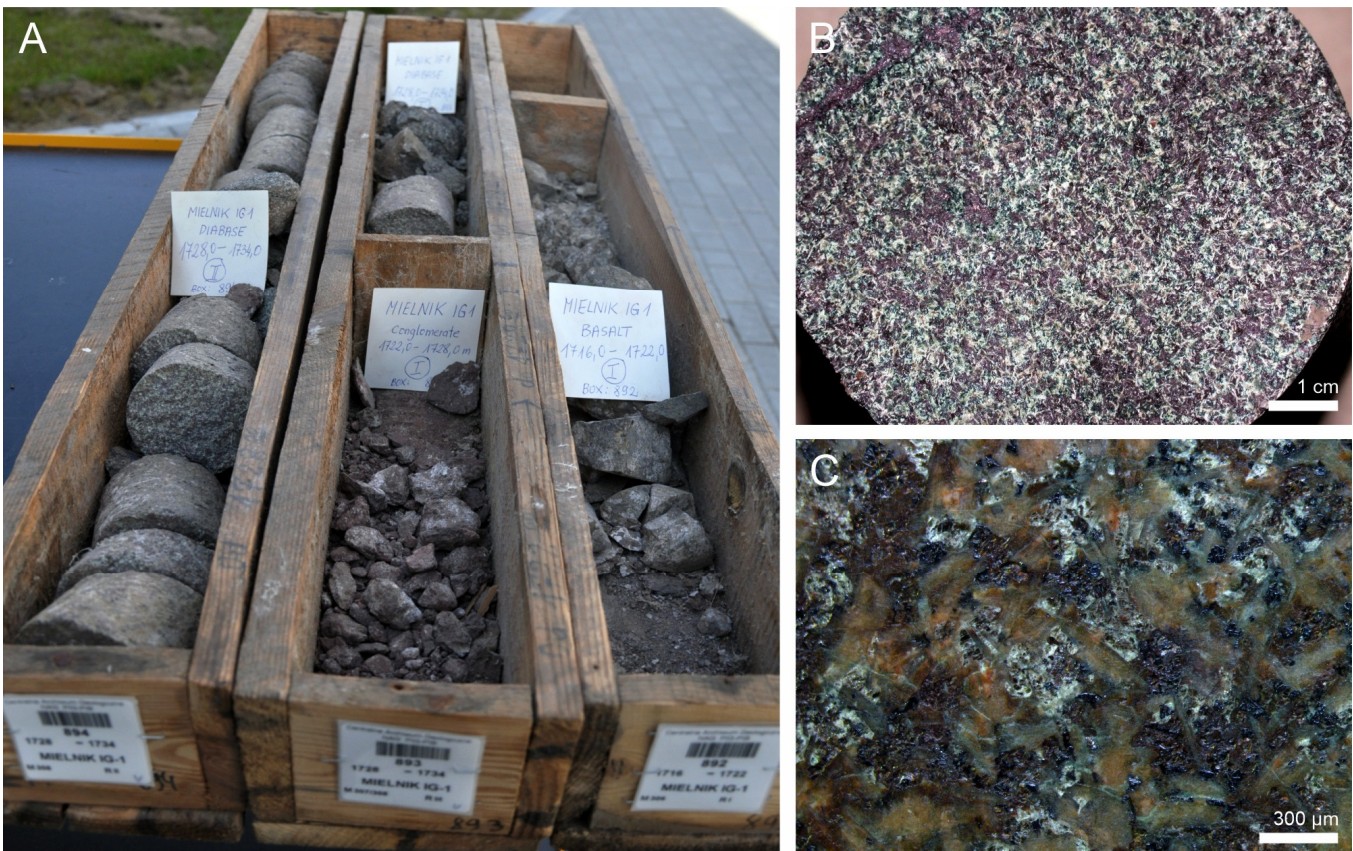

**Figure 4.** Photographs of the Mielnik IG1 drill cores from interval of ~1729–1722 m, showing: (**A**) Dolerite top (left), reduced interval of conglomerate of Żuków Formation (center) and basalts of Sławatycze Formation (right). The contacts were not retained; (**B**) Part of dolerite core sampled for U-Pb geochronology. Width of core is approximately 8.4 cm; (**C**) Dolerite rock chip (polished) to expose a poikilitic texture and main components coarse grains of plagioclase (albitized) and pyroxene and Fe–Ti oxides.

## 4. Methods

### 4.1. Geochronology

Zircon crystals were extracted from accessible drill core samples (Figure 4) collected from the bottom part of diabase section. Standard mineral separation procedures included crushing and pulverizing, followed by flotation through a magnetic separator, and flotation in heavy liquids to obtain a heavy mineral concentrate composed mostly of apatite and a few zircon crystals. The grains were hand-picked under a binocular microscope. All obtained zircons were glued on a double-sided tape. Zircon grains were mounted in epoxy, together with Temora 2 and 91,500 zircon standards, and ground to about half their thickness, and sequentially polished using 6 μm and 1 μm diamond suspension. All polished grains were imaged in reflected and transmitted light on Nikon ECLIPSE LV100POL microscope, Nikon, Tokyo, Japan, and in cathodoluminescence (CL) using the PGI-NRI, Warsaw HITACHI Su 3500 scanning electron microscope, Hitachi High-Technologies Corporation, Tokyo, Japan, with a CL detector.

Zircon was dated by the U–Pb method using the PIG sensitive high-resolution ion microprobe SHRIMP IIe/MC, following the methods described by Williams [53].

The U–Pb ratios were measured under the following conditions: intensity of the primary $O_2$–ion beam at 3.5 nA, with 20–23 μm spot size. For each analysis, six peak scans were collected for nine mass. Prior to each measurement, the surface of the analysis site was pre-cleaned by rastering of the primary beam for ~2 min., to eliminate surface common Pb. Obtained data was processed using the SQUID 2 program. U–Pb ratios were normalized to the TEMORA zircon standard (0.0668) corresponding to TIMS age of 416.78 Ma $\pm$ 0.3 Ma $^{206}$Pb/$^{238}$U age [54], accompanied with the reference zircons 91,500

used for elemental concentrations U = 82.5 ppm [55]. Analysis of the unknown zircon grain and the Temora 2 standard zircon were alternated every three analysis for the best control of Pb/U ratios. Measurements on Temora zircons (n = 17) from analytical session were coherent, yielding a concordia age of 416.7 ± 4.6 Ma (a lower intercept) or weighted average of $^{206}Pb/^{238}U$ age of 416.8 ± 4.2 Ma [1.0%] 95% conf. consistent with the ID-TIMS reference age [54]. Supplementary Table S1 shows the results, including contents of U, Th, and radiogenic Pb, as well as the U-Pb and ratios, measured in 32 zircon grains from the dolerite sample calculated according to SQUID 2 algorithms [56]. Zircon isotopic age and concordia diagram were calculated and drawn using Isoplot 3.0 [57] and isoplotR [58]. Measured compositions were corrected for common Pb using non-radiogenic $^{204}Pb$. The U-Pb zircon age data are presented in Table S1 (Supplementary Materials). Uncertainties given on single spot analyses are based on counting statistics and are at the 1 sigma level.

### 4.2. Mineral Chemistry

The mineral geochemistry was determined on polished thin section and polished rock chips that were selected from the studied interval of the dolerite sill.

The titanite was analyzed using the CAMECA SX100 microprobe, CAMECA, Gennevilliers Cedex, France, with a wavelength-dispersive spectrometer (WDS), but other mineral phases using the Oxford Instruments Energy-Dispersive Spectrometer (EDS), Oxford Instruments, High Wycombe, UK, coupled with a Leo Zeiss 1430 electron microscope, Carl Zeiss AG, Oberkochen, Germany, located at PGI-NRI. The standard operating conditions included an accelerating voltage of 15 kV, a beam current of 10 nA, with a beam diameter of 5 μm. Natural and synthetic standards were employed. Corrections were made using an on-line ZAF method. The results are reported in Supplementary Table S2.

### 4.3. Whole-Rock Geochemistry

The whole-rock analysis of the four Mielnik IG1 diabase core samples was performed at the Bureau Veritas Mineral Laboratories, former ACME Lab analytical laboratory, in Vancouver, Canada, according to the standard lithogeochemical LF100 and LF302 analytical packages. Preparation of samples for the geochemical study included lithium borate ($LiBO_2$) fusion and dissolution in acids. The major elements were analysed using standard Inductively Coupled Plasma Optical Emission Spectrometry (ICP-ES) techniques, while all trace elements were determined by inductively coupled plasma mass spectrometry (ICP-MS). The loss of ignition (LOI) was done gravimetrically by determining the weight loss of the examined samples after their roasting at 950 °C for two hours. The detection limits of major elements were close to 0.01 wt%, while the detection limits for most trace elements fill range from 0.1 to 1 ppm. The analytical data are reported in Supplementary Table S3.

## 5. Results

### 5.1. Geochronology

The morphology of crystals picked from the Mielnik IG1 dolerite, from a depth of 1745 m (Figure 5) suggests a dominance of inherited or xenocrystal forms, however several small and sharply faceted, prismatic crystals, indicated a primary igneous origin. The zircons from dolerite display a great variety of forms and textures, which are visible in both the transmitted-light (TL) and cathodoluminescent (CL) images (Figure 5E), with the grains providing evidence of similar fine-scaled simple oscillatory magmatic zonation. The rims or overgrowth were not recognized.

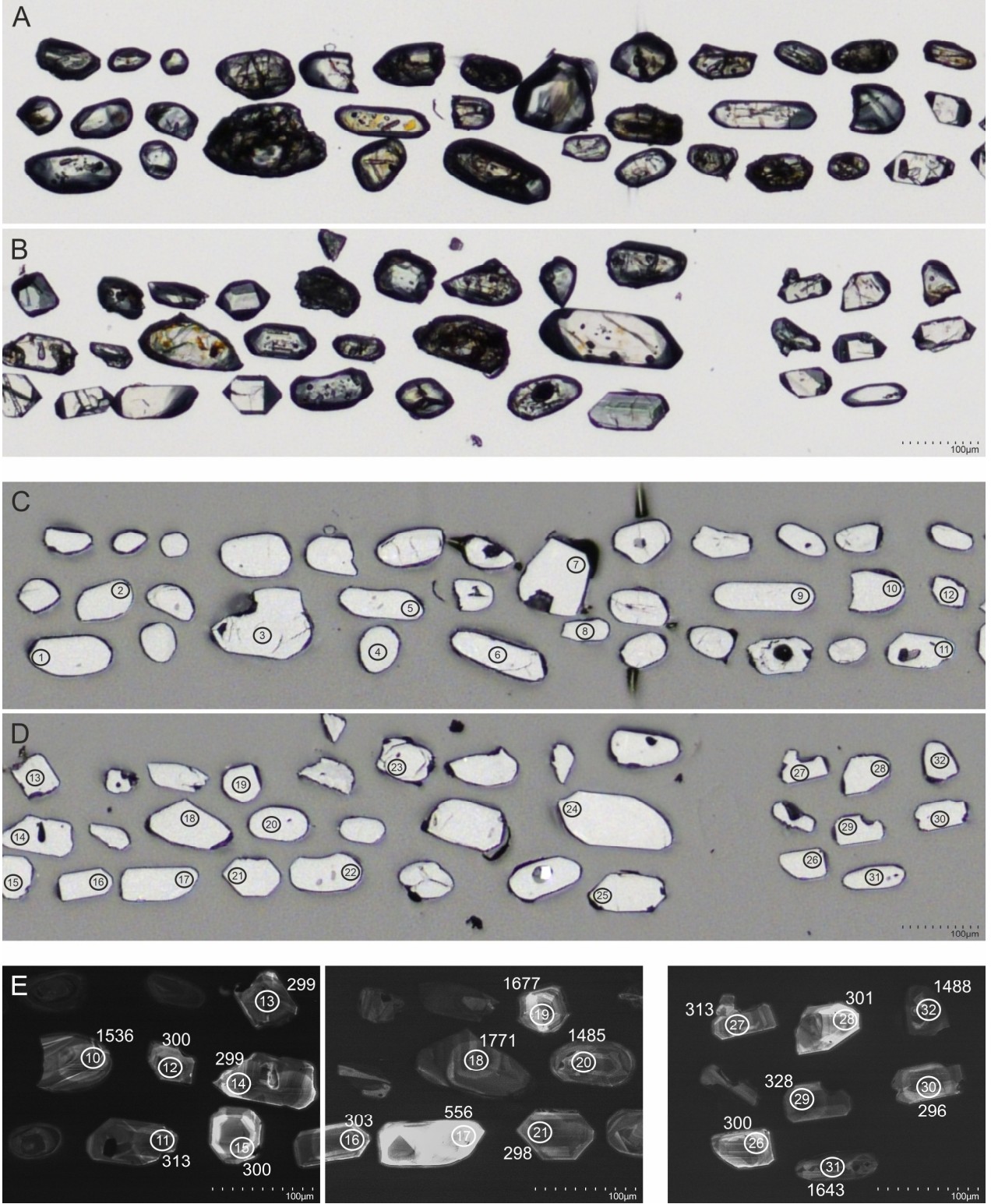

**Figure 5.** The zircon grains characteristics: (**A**,**B**) Transmitted and (**C**,**D**) reflected light photographs of the all crystals extracted from Mielnik IG1 dolerite, depth 1745 m; (**E**) selected cathodoluminescence images (CL). Circles represent the spot (diameter of 23 μm). positions. Halo after rastering (>23 μm) is visible. The analyses number or/and its age in Ma, according to Supplementary Table S1 Data.

Thirty-four spots on thirty-two crystals (including all prismatic preselected zircons) were analyzed (Supplementary Table S1, Figure 5C,D). Spot locations are marked on reflected-light (RL) images.

The obtained data shows a broad range of ages, from 2038 ± 39 Ma to 284 ± 8 Ma (Figure 6). More than half (58%) of the zircons provide significantly older ages than the rest of the analyzed grains e.g., Paleoproterozoic to Late Ediacaran (Figure 6B, Supplementary Table S1). They also show the heterogeneity of morphological form and diverse contents of Th, U and Pb trace elements (Figure 6D,G,H).

A separate group is formed by of Paleozoic results (n = 15/34), dominated by a cluster from the Carboniferous /Permian boundary with $^{206}$Pb/$^{238}$U ages ranging between 328 ± 6 Ma and 284 ± 8 Ma. They have consistent Th/U ratio in a narrow range between 0.32–0.46 (Figure 6F). These zircons contain Pb concentration primarily < 25 ppm, Th concentration < 200 ppm and U < 500 ppm. They form relatively coherent group (Figure 6F–H), suggesting a genetic relationship. In contrast to the older group the age results are mostly discordant, but there is one grain (14.1) with a relatively low level of discordance (+8% disc). The isotopic ratios were plotted on the Wetherill concordia diagram (Figure 6C) showing a linear regression with the upper intercept at 290 ± 11 Ma, and a mean square weighted deviation (MSWD) of 2.1. The selected youngest and most coherent results (n = 6 grains) yielded a concordia age of 299.6 ± 3.6Ma (MSWD = 0.021) and corresponding $^{206}$Pb/$^{238}$U weighted mean age of 299.5 ± 4.4 Ma (MSWD = 0.18) (Figure 6E). An alternative calculation for this youngest cluster has been made with $^{238}$U-$^{206}$Pb and $^{207}$Pb-$^{206}$Pb ratios as input using isoplotR [58] radial plot. Based on a larger number of analyzes (n = 11) a central age = 299.9 ± 3.3 Ma (MSWD = 0.32) has been obtained.

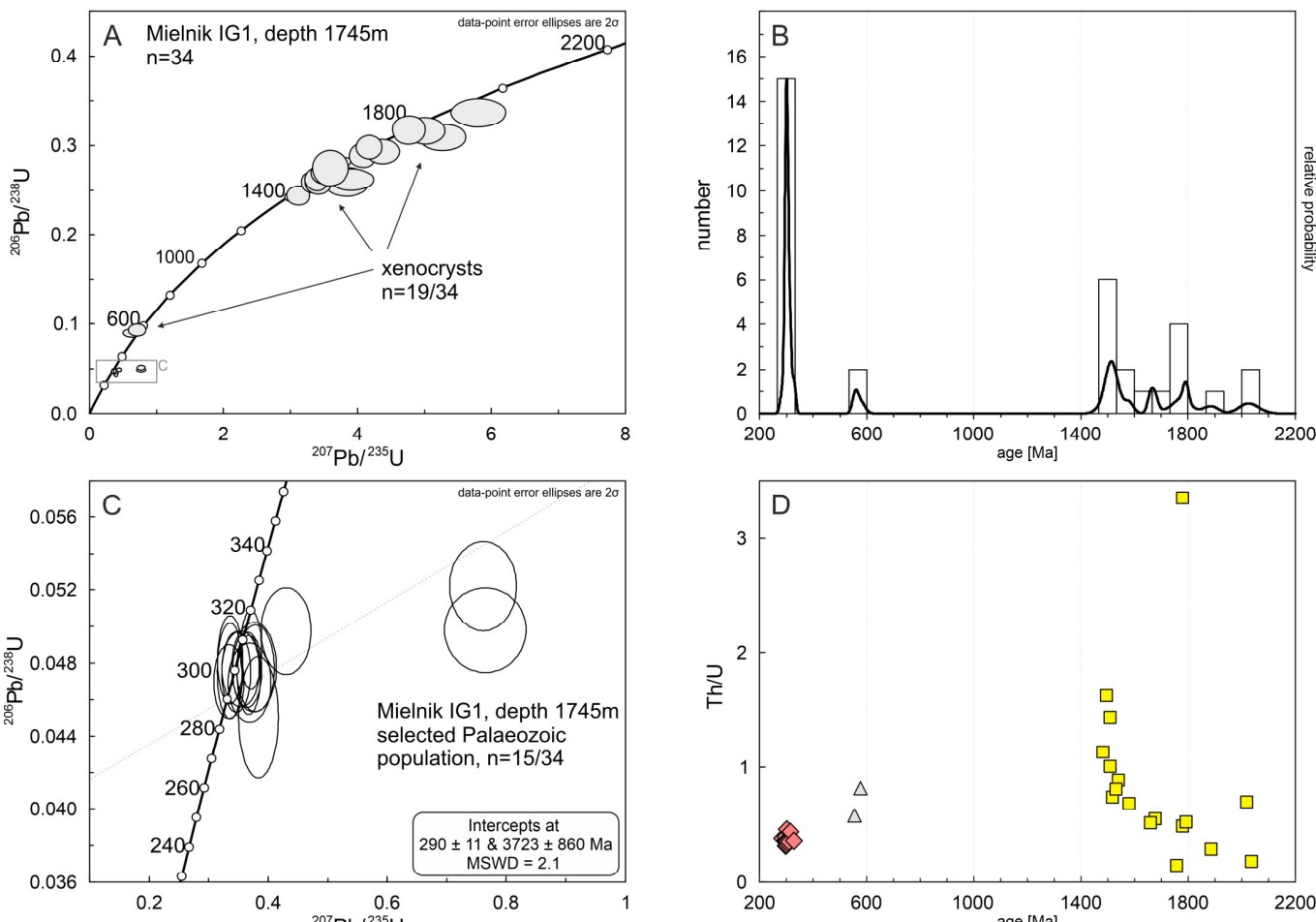

**Figure 6.** *Cont.*

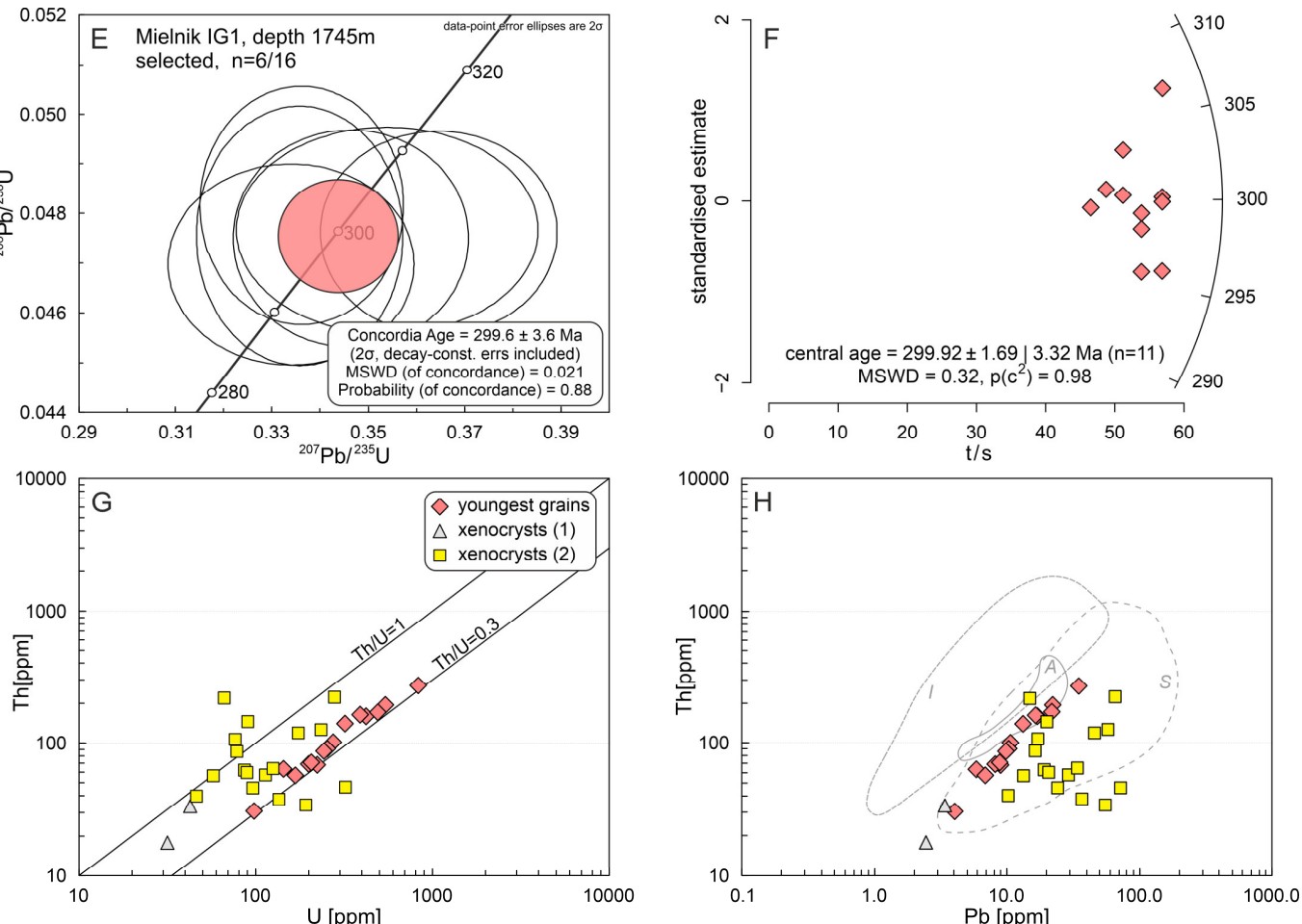

**Figure 6.** Zircon U–Pb age and trace elements diagrams for Mielnik IG1 dolerite: (**A**) Concordia plot of the all zircon data; (**B**) The histograms integrated with probability plots showing detected zircon populations; (**C**) Concordia age of selected Palaeozoic grains including discordant analyses; (**D**) Zircons Th/U ratios versus age plots showing a diversity of xenocrysts population (**E**) the Concordia diagram for selected youngest zircon cluster (n = 6); (**F**) alternative calculation of the youngest age using the radial plot based on isoplotR [58] based on U-Pb and Pb-Pb isotopic ratios; (**G**) Th and U contents defining a positive correlation within youngest zircon cluster; (**H**) plots of Th vs. Pb trace elements concentrations in zircon grains. The discrimination fields defined by zircon derived from I-, S-, and A-type granitoids [59] are shown for comparison.

## 5.2. Mineral Chemistry

Plagioclase and clinopyroxene are, or were, the most abundant minerals in the Mielnik IG1 dolerite. They were accompanied with subordinate olivine (Figure 7). Prominent accessory phases include magnetite, and minor titanite. Other accessory minerals include apatite and very rarely zircon.

Clinopyroxene from the Mielnik IG1 dolerite (two samples from the bottom of the sill) are plotted on the Morimoto [60] classification diagram (Figure 8A). Their compositions fall mostly in the diopside field and on the edge of augite field, forming a continuous trend of ranges: $Wo_{42–46}$ $En_{33–41}$ $Fs_{16–21}$. All EDS analyses document a titanium contents from 0.7 to 2.5 wt% $TiO_2$. The $Al_2O_3$ concentrations are variable (0.67–3.4 wt%), suggesting a change in pressure during melt rise. To estimate crystallization temperature (T) or pressure (P), two multivariate diagrams of Soesoo [61] were employed. This graphical empirical method indicates a crystallization P below 2 kbar (Figure 8B) and T close to 1100 °C (not shown).

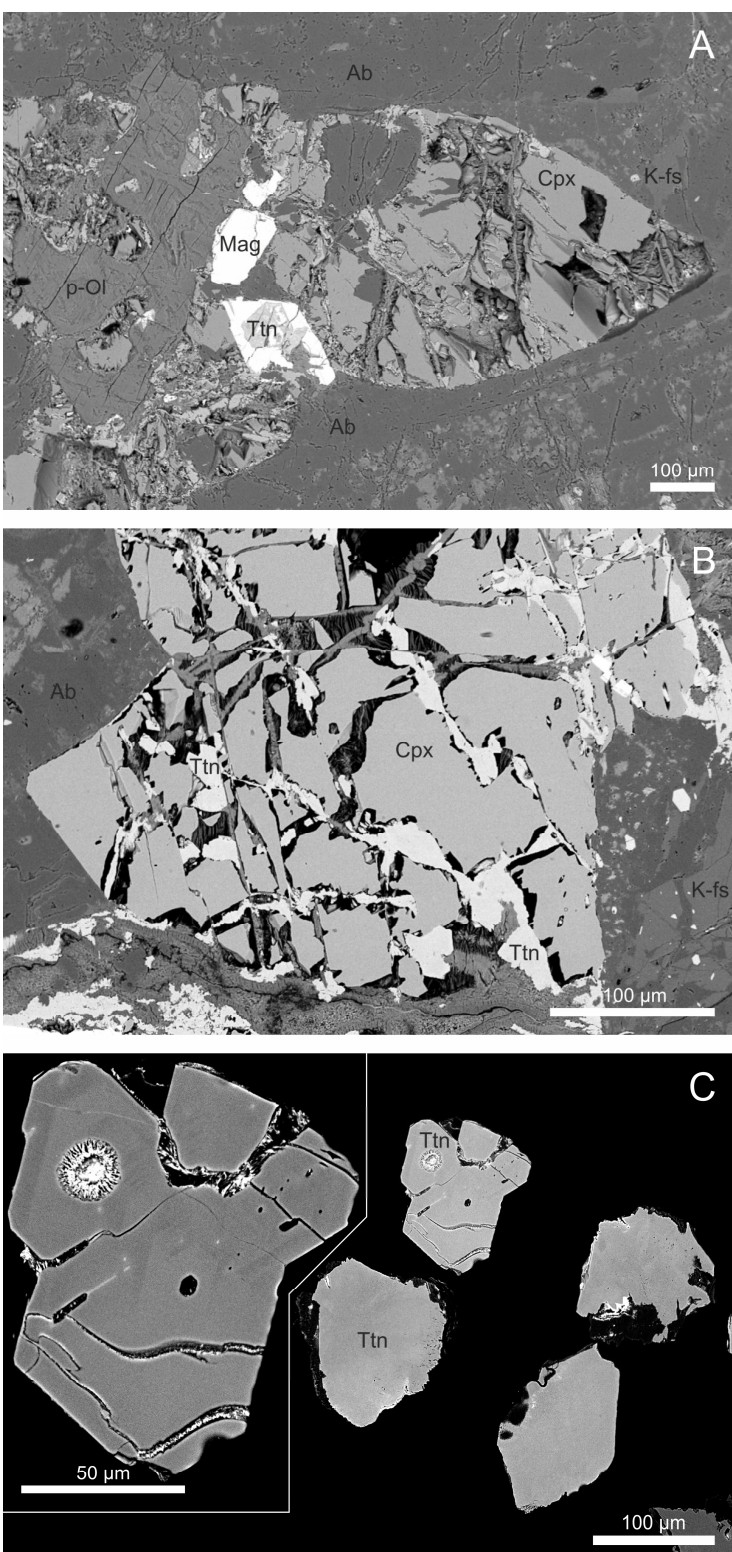

**Figure 7.** Petrological characteristics of the Mielnik IG1 dolerite samples of 1746 m and 1745 m: Back scattered electron BSE images showing: (**A**) the igneous texture with fine-grained of clinopyroxene and magnetite and albitized plagioclase; Grains of clinopyroxene (Cpx), ilmenite (Ilm), alkali feldspar, magnetite (Mgt), plagioclase (Pl) and quartz (Qtz) and titanite (Ttn) are indicated. (**B**) The clinopyroxene of diopside/augite composition with of magnetite and titanite grains; (**C**) morphological types of titanite grains. The single crystals extracted from heavy mineral concentrate after zircon separation and analyzed by EPMA (Supplementary Table S2).

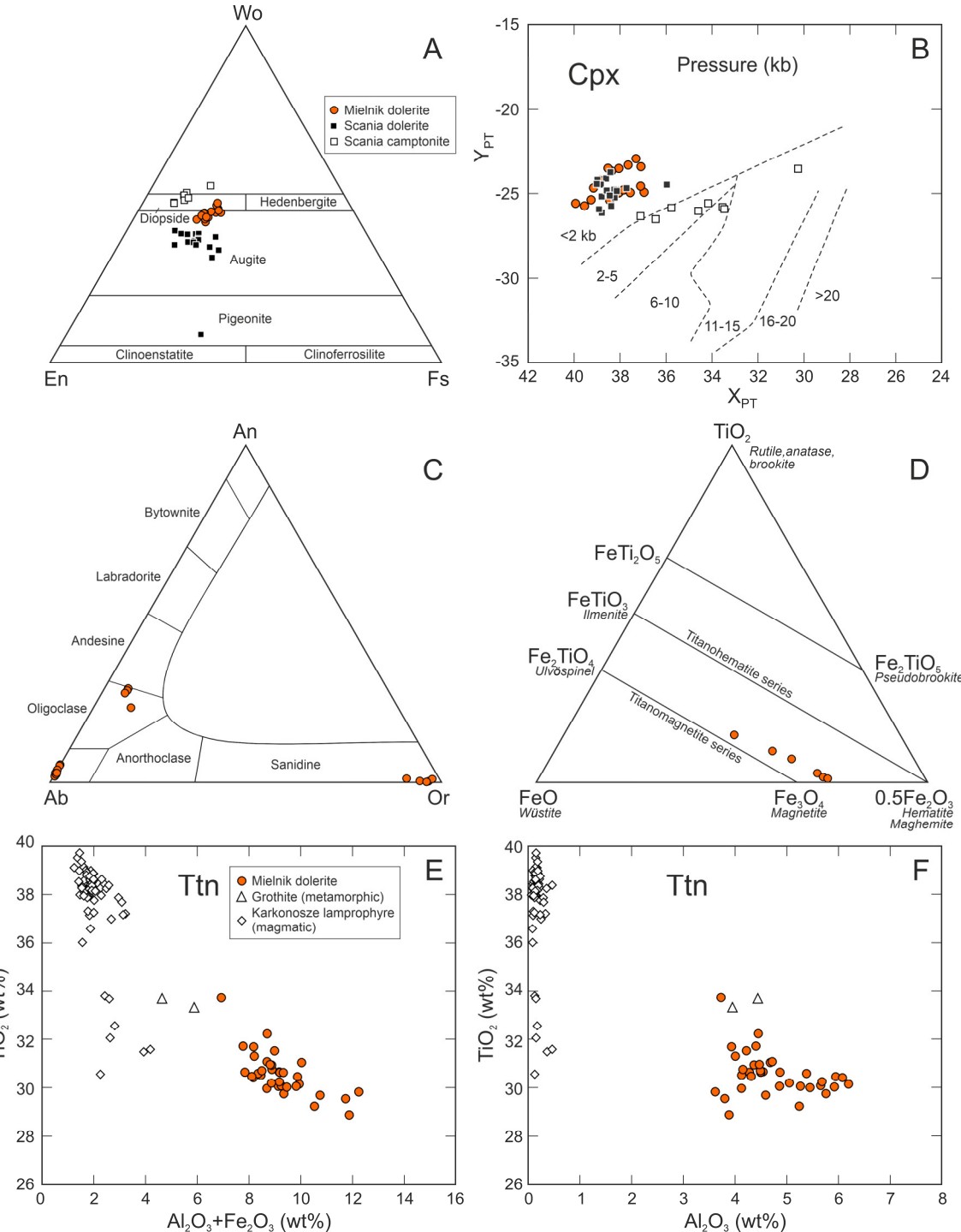

**Figure 8.** Diagrams showing the chemical composition of mineral phases from the Mielnik IG1 dolerite (**A**) Wo-En-Fs classification diagram for clinopyroxenes [60]; (**B**) An empirical coordinates for estimations of a pressure values calculated from the clinopyroxene composition [61]; (**C**) Composition of feldspars projected on triangular diagram of albite (Ab), anorthite (An) anorthoclase (Or) end-members; (**D**) Composition of Fe oxides projected on triangular $TiO_2$-$FeO$-$Fe_2O_3$ solid system diagram; (**E**) Variations of titanite in $TiO_2$ vs. ($Al_2O_3$ + $Fe_2O_3$), indicating a scope of substitution on the titanite Ti site (either $Al^{+3}$ or $Fe^{+3}$); (**F**) Titanite $TiO_2$ vs. $Al_2O_3$ variations controlled by pressure conditions [62]. The composition of titanite close to the stoichiometric was taken from the plutonic Karkonosze Mt lamprophyres [63] and metamorphic grothite Al–Fe rich [64].

The dolerite contains mutual intergrowths between pyroxene and olivine (Figure 7A). Olivine is considered to be one of the least stable silicate mineral. The BSE images and EPMA analyses show, that all olivine crystals are completely altered. An approximated composition of this product has been analyzed by EDS as being $SiO_2$ = 33.4–34.1wt %, MgO = 17.8–18.9 wt%, $Fe_2O_3$ = 18.1–19.6 wt% and $Al_2O_3$ = 11.3–12.2 wt%. Throughout the alteration process and the deuteric cations movements in olivine, there is a decrease in $SiO_2$, FeO and MgO and enrichment in Al-contents that were observed.

Plagioclase crystals in the Mielnik IG1 dolerite have undergone albitization (Figure 7A). EDS analyses of the feldspar domains indicate a variable mixture, mostly of albite and orthoclase (Figure 8C). The plagioclase composition has been significantly modified. Less altered fragments indicate the presence of the oligoclase (20–27% An).

Fe- oxide analyses show a variable $TiO_2$ content, ranging from 1.8 to 18.0 wt%. All plots (Figure 8D) are located along line of the titanomagnetite series on the $FeO$–$TiO_2$–$Fe_2O_3$ ternary diagram, falling within the titanohematite field. It indicates a deuteric oxidation and hematitization processes. A low content of MgO always below 0.5 wt %, (e.g., low geikeilite component) suggests late-stage crystallization or a secondary alteration processes.

Accessory titanite occurs among clinopyroxenes and Fe-Ti oxides, forming irregular-shaped grains (Figure 7B,C). Several WDS analyses show elevated contents of $Al_2O_3$, ranging from 3.62 to 6.20 wt%. The titanite crystals also contain high amounts of iron in the range from 2.97 to 8.62 wt% $Fe_2O_3$. Most of the analyses (n = 24/36), show that Fe predominates over Al. The ratio Fe/Al ranges between 1.1 and 2.7, that is characteristic for silica-undersaturated igneous rocks [65].

The increase in the sum of $Al_2O_3 \pm Fe_2O_3$(total) at the expense of $TiO_2$ reaches values between 7–12% (Figure 8E,F), which deviate from the stoichiometric composition uncommonly observed in magmatic titanites e.g., lamprophyres [63]. Morad et al. [66] suggested that the coupled $Al^{3+}$, $Fe^{3+}$ substitution in the Ti-site results from calc-silicate alteration, where $Al^{3+}$ is derived from adjacent plagioclase and $Fe^{3+}$ from adjacent magnetite. This case fits perfectly into the mineral assemblage of the investigated dolerite. Apart from prominent Al and Fe substitutions, titanite from the Mielnik IG1 dolerite always has an increased content of vanadium, yttrium and zirconium (Supplementary Table S2), in ranges of V: 660–2320 ppm, Y: 110–320 ppm and Zr up to 670 ppm (with EPMA detection limit for V: 73 ppm, Y: 104 ppm, Zr: 142 ppm). The $V^{5+}$ substitution in titanite on the Ti-site suggests oxidizing conditions above the hematite-magnetite buffer during calc-silicate alteration [67], which fits perfectly into the context in the Mielnik dolerite. In general the presence of the titanite tightly constrains the stability of mineral assemblage. For an iron-rich composition reaction between titanite, Fe-Ti oxides, clinopyroxene, olivine and quartz, one must constrain the P-T—oxygen fugacity conditions [68].

The incorporation of Zr into titanite has been found to be strongly dependent on temperature [69]. The measured maximum contents indicate the crystallization temperature did not exceed 740 °C. Titanite is known to incorporate increasing amounts of Al with increasing pressure. Testing a new empirical barometer formula [62] values between 4.4 and 6.6 kbar were calculated. These are values higher than those calculated on the basis of the clinopyroxene composition (Figure 8B).

### 5.3. Major and Trace Element Whole-Rock Geochemistry

The samples from the Mielnik IG1 dolerite taken from different depths show low silica content in the range of 45.50–49.59 wt% $SiO_2$, moderate $TiO_2$ in the range of 2.03–2.33 wt%. These samples are characterized by high $Al_2O_3$ contents in the range of 14.99–15.18 wt%, and $Fe_2O_{3T}$ in the range of 12.28–14.27 wt% and MgO concentrations in the range of 5.26–7.80 wt% with Mg# values from 50.0 to 52.1 wt% (Supplementary Table S3). The CaO content is very low in the range of 2.89–3.65 wt%. The sum of alkalies ($Na_2O$ + $K_2O$) ranges from 6.0 to 6.96 wt%. The dolerite samples revealed relatively high LOI, from 4.20 to 6.00 wt% that correspond to a high proportion of hydrous alteration minerals (olivine pseudomorphs). Despite clear evidence that all samples in this study are altered, these

modifications of mineral and whole-rock chemistry is less concerned with the content of immobile trace elements.

The Mielnik IG1 dolerite is classified (Figure 9A) as a sub-alkaline basalt on the Winchester and Floyd diagram (1977). The dolerite is characterized by elevated contents of Rb (38.9–65.3 ppm), Ba (617–869 ppm), and Sr (294–403 ppm), but low contents of Th (1.0–1.3 ppm), Nb (9.6–11 ppm) and Zr (110–127 ppm), Y (22.8–30 ppm).

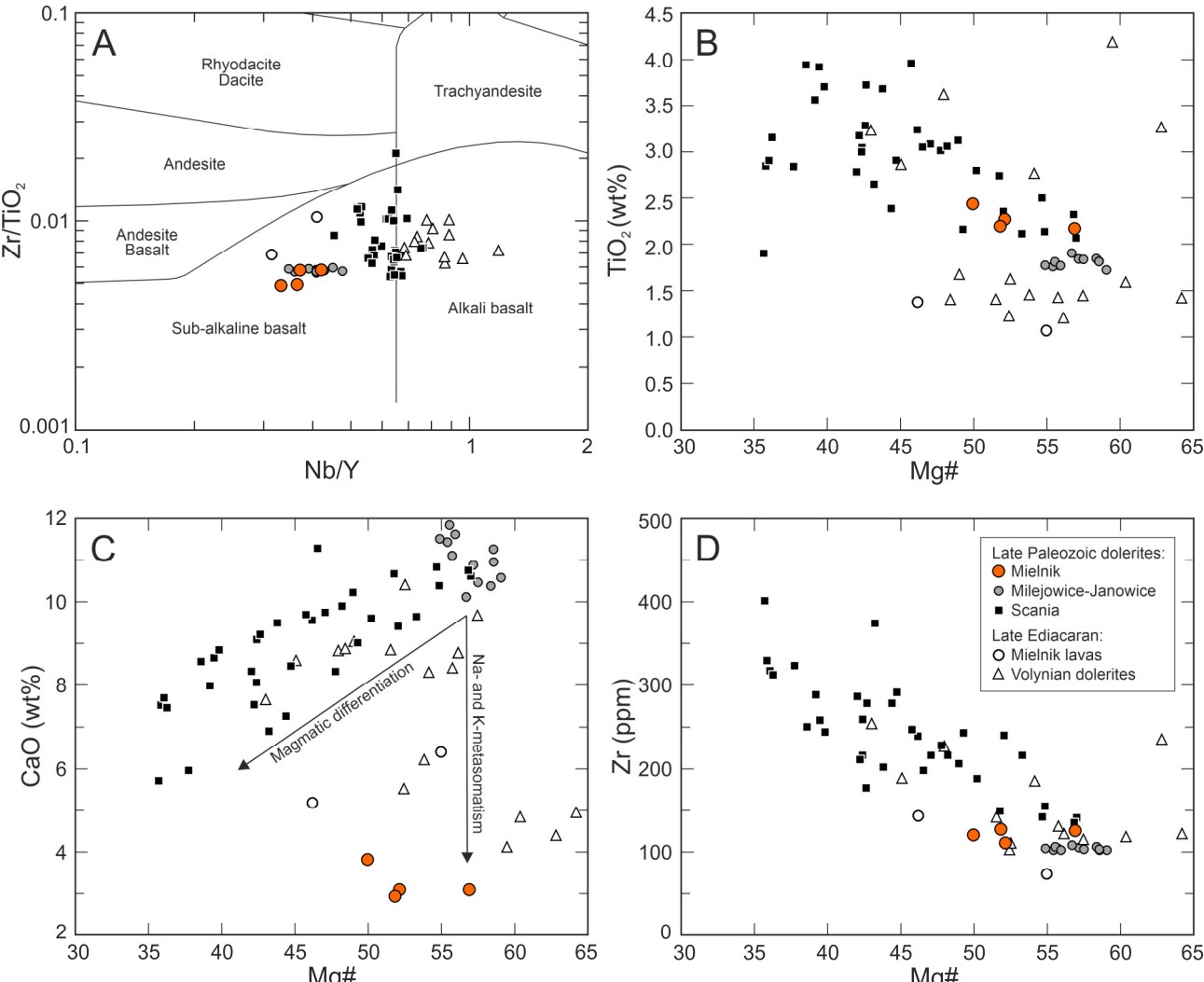

**Figure 9.** Geochemical characteristics of Mielnik IG1 dolerite (**A**) Winchester and Floyd [70] classification diagram; (**B**–**D**) Bivariate diagrams using mg-number (Mg#) against selected major and trace element oxide composition: CaO and TiO$_2$ vs. Zr. Data characterized other dolerite from European magmatic provinces are shown for comparison: Scania dolerite dykes: [9]; Milejowice Janowice dyke of Holy Cross Mt: [71]; late Neoproterozoic: dolerite dykes from Volyn [25], basalt from Mielnik IG1 depths 1715 m, 1695 m from [24].

All analyzed samples have moderately elevated total rare earth element (REE) contents (115–124 ppm). They display a similar REE pattern (Figure 10A) showing an enrichment in light rare earth elements (LREE) with (La/Yb)$_N$ ratios of 4.84–6.08 (Supplementary Table S3) and no europium anomaly (Eu/Eu* = 0.99–1.10). The heavy rare earth elements (HREE) pattern with (Gd/Yb)$_N$ ratios of 1.64–1.95 points to similarities to the type of characteristics of an enriched mid-ocean ridge basalt (E-MORB). On the primitive mantle-normalized (PM) multi-element diagram, the studied mafic rocks exhibit strong enrichments in Rb, K, and Pb, but slight depletions in Th, Nb, and Zr.

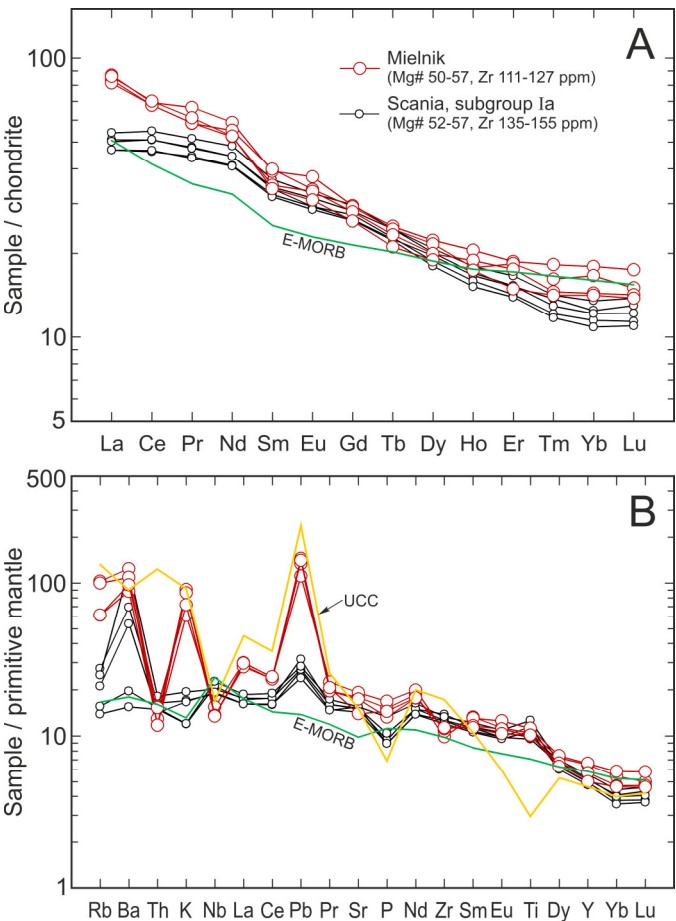

**Figure 10.** (**A**) chondrite-normalized rare earth element compositions (**B**). Primitive mantle-normalized multi-element diagrams, including comparison to selected dolerite from Scania [9]. The normalizing values are from [72]. The enriched mid-ocean ridge basalts (E-MORB) and upper continental crust (UCC) patterns are shown as a frame of reference. Average values taken from [72,73].

## 6. Discussion

### 6.1. Age Interpretation

Zircon is rather unique phase in mafic rocks such as basalts because its crystallization from low silicate melts requires a condition the oversaturation of both $ZrO_2$ and $SiO_2$ [74–76]. Nevertheless, zircon can be encountered in mafic plutonic rocks, where it have crystallized from late stage and more evolved melts. Therefore from an initially undersaturated melt the precipitation of zircons only seems possible by decreasing solubility due to cooling or/ and by evolving melt composition. In case of mafic dolerite from Mielnik IG1 a significant crustal contamination documented by whole rock geochemistry (Figure 10B) could be responsible for the origin of oldest or even most of zircons. Hence an interpretation of the nature of the youngest grains is crucial.

The youngest group of zircons present a features indicative of a simple magmatic origin with weak oscillatory zoning. There are no elongated crystals, more characteristic for volcanic rocks. The youngest grains from subvolcanic rock in Mielnik IG1 are very fine. They occasionally preserve euhedral crystal faces (Figure 5). This has been attributed to the fact that zircon is a late crystallizing phase, from more evolved melt potion. These grains undoubtedly crystallized from the same melt. They have almost uniform Th/U ratio and positive corelation between Th versus U and Pb contents (Figure 6D,G,H). These trace elements concentrations exposes a very similar trend as has been defined by composition of zircons hosted in A type magmatic rock (granitoids) in contrast to plots of the older xenocristic grains, that better correspond to field of S-type magmatism [59]. This method

of discrimination was dedicated to a detrital zicons, but allows to expose the uniform characteristics of the youngest group. Such a uniform group does not prove their xenocrystic nature. Presumably they were formed at a late stage of igneous crystallization of dolerite in Zr-enriched interstitial melt pockets. Their crystallization age tentatively is interpreted as close to the age of intrusion emplacement.

The remaining zircon grains (n = 19/34), defined as a truly xenocrysts, make up a majority of available zircon grains. Their age mainly falls in the range of 1480–2038 Ma, and may reflect a few distinct, but relatively local sources. The contamination with Late Ediacaran volcanogenic zircons was subordinate (n = 2). The age of xenocrysts (Figure 6B) corresponds with detrital zircon populations commonly obtained from the Ediacaran or Early Paleozoic clastic rocks observed in the Mielnik IG1 section (Figure 3) at depths of 1540 m, 1582 m, 1628 m [45] or in the Late Ediacaran pyroclastics at a depth of 1710 m [7]. The xenocrysts were incorporated just before emplacement of dolerite; they were not resorbed.

A comparison of the geochemical and U-Pb age data obtained from the Mielnik IG1 dolerite, on a wider regional basis, has revealed that the Late Carboniferous dykes are unique in Eastern Fennoscandia region. Only the dyke groups located in vicinity of early Carboniferous alkaline-mafic massifs (Ełk, Tajno, Ciechanów) were known (Figure 2).

Geophysical and geological data documents a complex tectonic evolution of the EEC's margin and its cover during the Paleozoic time. A series of reverse faults in the upper crust have been reactivated many times as a result of transferring stress in the lower ductile crust undergoing buckling to the upper crust [77]. Apart from the faulting, a dominant feature of the cratonic edge remains an abundance of a dolerite and (mafic) dyke swarms intruding both, the EEC basement and the lower Paleozoic platform sediments [9].

### 6.2. Paleotemperature Evidence

A paleothermal investigation of the platform sediments performed recently, using multiple low-temperature thermochronometer methodologies e.g., apatite fission-track (AFT) and zircon (U-Th)/He (ZHe) and a K-Ar analysis of the clay fractions [78–80], allows for the identification of a few thermal events, which were related to the area of cratonic foreland of its neighboring orogenic systems. K-Ar dating of the clays from volcanogenic rocks (E Poland and W Belarus) documents [78] a thermal activity event coinciding with the events of the Caledonian orogeny (dated at 417–447 Ma). The second pulse of thermal activity was related to the Late Variscan time period dated ca. 300 Ma (op. cit.) was identified more to the south, close to Volyn area. Moreover, thermal modeling [80] showed that significant heating of the Ediacaran and Carboniferous sedimentary successions occurred before the Permian, with maximum paleotemperatures occurring in the earliest and latest periods of the Carboniferous for Baltic-Podlasie and Lublin Basins, respectively [80].

The Lower Paleozoic rocks in almost all the Polish parts of the platform sediments reached maximum paleotemperatures in the Carboniferous between 320–340 Ma [79], which was related to the emplacement of at least three prominent alkaline- ultramafic massifs (Figure 2) that intruded between 348–338 Ma [31,33]. Moreover, in the southern part of the Lublin Basin (Kaplonosy, Łopiennik) and in the Łysogóry Unit of Holy-Cross Mts., the maximum paleotemperatures were related to the Early Permian and range between 290–270 Ma (K-Ar age), in response to volcanic activity of the same age, occurring in the Kraków-Lubliniec Fault Zone (KLFZ).

Such evidence of various magmatic and tectonic activities (regional uplift) jointly reflect the thermal perturbation of the lithosphere. The elevated Permian heat flow was probably a consequence of Early Permian continental rifting [80].

### 6.3. Coeval Age of Dykes' Emplacement along Edge of the EEC

The thousands of WNW- to NW-trending dykes of various thicknesses and lengths in Scania [9], Västergötland and Rügen [11] revealed a Permo-Carboniferous age defined by either, the Ar-Ar or K-Ar methods [11,37,81].

The similar age of emplacement has been obtained from zircon autocrysts of the Mielnik IG1 dolerite sill, that is located further to the southeast within the structure of the Late Ediacaran aborted rift [6,7] dominated by the Volyn series volcanics (Figure 1). The occurrence of the Permo–Carboniferous dyke at the same depth of the Mielnik IG1 borehole documents an area of crustal stress, and a regional scale extension accompanied by a short pulse of magmatic activity.

A dominant age of 298–300 Ma (n = 54 samples) obtained from the dolerites in Scania (Skåne) by using the K-Ar method on plagioclase, biotite and pyroxene fractions [37], corresponds well with the concordia age of 300 Ma obtained by using the U-Pb method on autocrystic zircon from the Mielnik IG1 dolerite (Table 1). Despite the various methods to determine the age (K-Ar versus U-Pb) these results can be considered as coeval. In case of veins of sub- volcanic rocks, where a cooling of the magma is rapid, the geochronological systems exhibiting relatively low closure temperatures (~300 °C) can be efficiently used to date an emplacement time. They may yield results indicating an age comparable (within an error) to those provided by a high temperature geochronometers [82]. In the case of Scania dyke swarm geochronology, the K-Ar results [37] were confirmed by preliminary U-Pb baddeleyite dating of the dolerite yielding an age of 297 ± 10 Ma and Ar-Ar ages clustered ca. 305–298 Ma [81]. The dates found for the Scania older pulse and the Mielnik IG1 dolerite agree well with the timing of initial magmatism in the Oslo Rift [83] bracketing narrowly between 300.4 ± 0.7 Ma and 298.9 ± 0.7 Ma.

*6.4. Geochemical Comparison of the Dolerites*

It was demonstrated by Neumann et al. [13], that the Late Carboniferous-Permian extensional magmatism expressed by dyke swarms, with activity concentrated in a narrow time-span from ca. 300 to 280 Ma, was distributed unevenly in northern Europe, between North England, the Oslo Graben in Norway, the Scania of southern Sweden, all the way to Rügen in northern Germany and the Danish island of Bornholm. Their geochemical signatures reflect distinct magma sources [84]. To define possible similarities, the available geochemical data from a few significant dolerite-rich regions in the vicinity of TTZ, were compiled. Apart from the Late Carboniferous dolerites from Scania [9], the coeval dykes from Janowice and Milejowice [41,71], in the Łysogóry Unit of the Holy-Cross Mts., were also compared (Figure 9). They have been identified as a post-tectonic "diabase" formed during an extension on the Baltica passive margin.

Since the the Mielnik IG1 dolerite has thus far been considered a part of the Late-Ediacaran Volyn Series, the geochemistry of the Volyn dolerites and selected lavas from the Mielnik IG1 drill section, depths 1715 m, 1695 m [24], were taken into consideration.

The Milejowice and Janowice HCM (abbr. from Holy-Cross Mts.) dolerites represent a sub-alkaline basalt with similar $TiO_2$ and Mg# content (Figure 9), however the Mielnik IG1 samples have been strongly altered. These changes proceeded to occur in an oxidizing environment, evidenced by the increasing amount of hematite and $Fe^{3+}/Fe^{2+}$ ratio values of the bulk minerals, and resulted in the loss of Ca (dissolution of plagioclases), compensated by a gain in K and Na concentrations. This is in contrast to the rocks from Milejowice and Janowice, where the original composition of the olivine was preserved [71].

Binary variation diagrams of a few major and trace elements demonstrate the existence of at least two groups of dykes in Scania [9]. Based in this, the Mielnik IG1 dolerite remains a more mafic component (Figure 9), rich in Mg and poor in Zr, but it does fit well with the plots sequence. The chemical composition of pyroxenes and their crystallization conditions are relatively similar.

The multi-element diagrams normalized to chondrite and primitive mantle (PM) (Figure 10) reveal the slight differences of a pattern with a slight enrichment in LREEs, compared to Scania dolerites. The Mielnik IG1 dolerite has a more fractionated LREE pattern e.g., $(La/Sm)_N$ = 4.84–6.08 (Figures 10A and 11B, Supplementary Table S3), but HREE fractionation with values $(Tb/Yb)_N$ = 1.38–1.61 lower than those from Scania, ratio $(Tb/Yb)_N$ = 1.7–2.0, that suggest its formation at shallower levels.

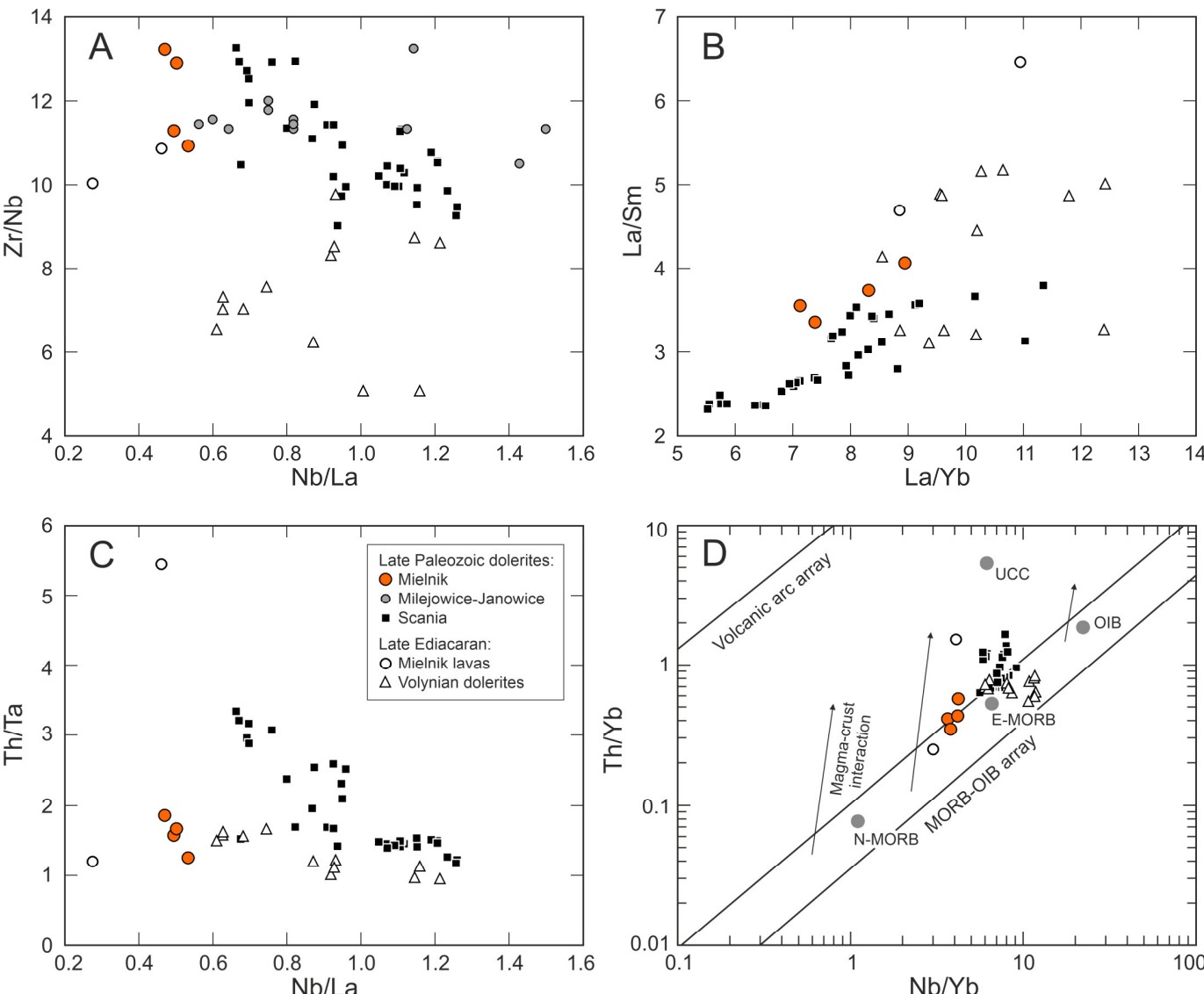

**Figure 11.** A comparison of the whole-rock chemical composition of Mielnik IG1 dolerite with that of other Carboniferous/Permian dyke swarm and Late Neoproterozoc dolerite and lava including previous analyses of [9,24,25,71] for (**A**) Zr/Nb versus Nb/La (**B**) La/Sm versus La/Yb, (**C**) Th/Ta versus Nb/La (**D**) Th/Yb versus Nb/Yb.

The elevated concentrations of the mobile elements, including Rb, Ba, K positive anomalies (Figure 10B) reflect a post-magmatic alteration processes that changed of the primary mineral composition (albitization, K-metasomatism).

The negative anomaly of Zr found in the PM normalized plot (Figure 10A) is indicative of a mantle source since Zr is considered to be a relatively immobile element [70]. The variations of the Zr/Nb ratio used to define a source's heterogeneity, reveal differences to the Volyn dolerites, but points to some similarities with the local flood basalts from the Mielnik IG1 drill section (Figure 11A). Almost-similar values are observed in the coeval Janowice and Milejowice HCM dykes [71].

The trend of increasing Zr/Nb and decreasing Nb/La may reflect a distinct local crust, as well as contamination. This is also suggested by the plot (Figure 11B) involving Th/Ta v. Nb/La.

The Th/Yb versus Nb/Yb diagram (Figure 11D) compares the different tectonic regimes [85] and is used for the detection of crustal signatures and to differentiate between the Mielnik IG1 and Scania dolerites. The plots displaying separate groups, in close proximity to the E-MORB end-member, along the boundary of the mantle array, however

both have had to been affected by crustal contamination. In the Mielnik IG1 dolerite a magma-crust interaction has been documented using zircon xenocrysts.

In general, these geochemical characteristics show a certain amount of variation in terms of their major and trace element concentrations, which could possibly be due to mantle heterogeneity and the influence of crustal contamination. A similar conclusion can be drawn from the compilation of previously obtained isotope data (Figure 12).

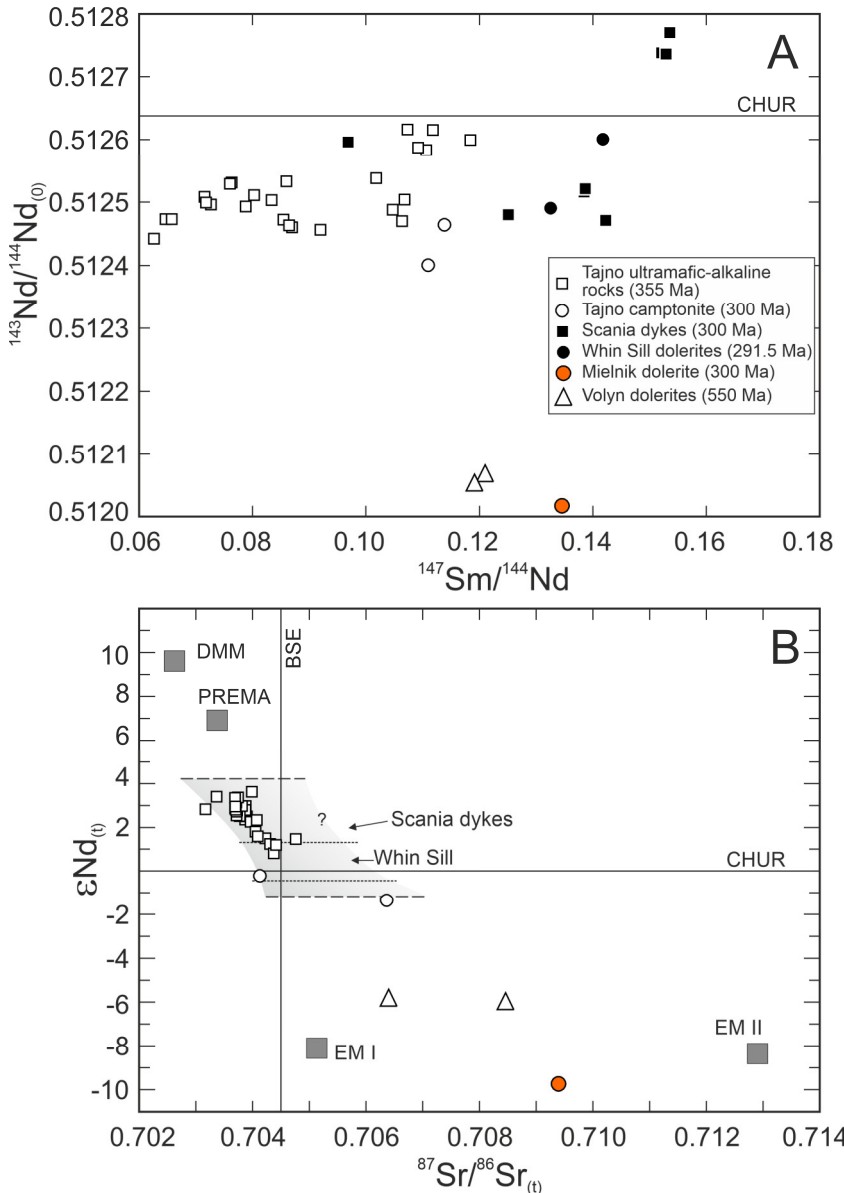

**Figure 12.** Compilation of published Sr–Nd isotopic data on the plots (**A**) $^{143}Nd/^{144}Nd_{(0)}$ ratio against $^{147}Sm/^{144}Nd$ ratio and (**B**) $ENd_{(t)}$ against 87Sr/86Sr, including the Carboniferous/Permian dyke swarms from Scania, Whin Sill of Northern England [9,86], data from Tajno alkaline ultramafic suite [30,87] and Mielnik dolerite dyke (recalculated after age revision) and Late Neoproterozoic dolerite of the Volyn igneous province [25,88] The mantle components DMM (depleted MORB mantle), PREMA (prevalent mantle), and EM I (enriched mantle, type I) are taken from [89], EM II (enriched mantle, type II) from [90].

The comparison of published Nd and Sr isotope ratios from a few Carboniferous intrusions collected in Figure 12 shows an individual character of the Mielnik IG1 dolerite. This compilation includes data from Scania dolerites [9]), Whin Sill of Northern England [86]

together with data from TTZ margin of the EEC with distinct ultramafic−alkaline to carbonatite rocks of the Tajno massif [30,87] and the dolerites from the Volyn province and recalculated result of the Mielnik IG1 dolerite [25,88]. The variations in $^{143}Nd/^{144}Nd$ (Figure 12A) demonstrate at least two separate type of sources. The low value of $^{143}Nd/^{144}Nd$ of the Mielnik IG1 sample is undoubtedly related to the contamination of its parent melt with the material of the continental crust. The impact of contamination is demonstrated by significant similarity of the Mielnik IG and UCC patterns on multielement diagram normalized to PM (Figure 10B). It is the contrast to the most of the Tajno rocks including coeval (300 Ma) camptonite, that plot in the depleted mantle quadrant of the Nd-Sr correlation diagram (Figure 12B) with low $^{87}Sr/^{86}Sr$ ratios and positive $ENd_{(t)}$. The higher values of $^{87}Sr/^{86}Sr$ ratios = 0.70940 and strong negative $ENd_{(300)} = -9.75$ obtained in Mielnik IG1, and that plot in the enriched mantle quadrant, rather exclude the same source and the petrogenetic connection with early Carboniferous Tajno suite. Most likely the dolerite remains a more individual response to the regionally intensified extension.

## 7. Conclusions

The U-Pb zircon age investigation of the Mielnik IG 1 dolerite from cratonic margin of the EEC provides more evidence documenting the extent of influence of the Permo-Carboniferous extension and rifting, and accompanied by magmatic pulses that were widespread across the area of extensional models [13]. The youngest zircons dated an evolved portion of magma at the late stage crystallization at $300 \pm 4$ Ma.

The Mielnik IG1 dolerite age and the older pulse of the Scania dyke swarm dates agree well with the timing of initial magmatism in the Oslo Rift [83] bracketing narrowly between 300.4 and 298.9 Ma, and a baddeleyite U-Pb age of $297.4 \pm 0.4$ Ma for the Great Whin Sill Dolerite Complex, N.E. England [36].

It seems plausible, therefore, that some of these dolerite and dyke swarms could have been emplaced during a relatively brief pulse of mafic magmatic activity, over a timespan of only a few million years, but over a distance of several thousand kilometers.

This specific magmatism was generated at different depths and was associated with distinct crust and mantle interactions, which is reflected in the geochemical and isotopic signatures.

The Mielnik IG1 dolerite can constitute an important time marker in the regional stratigraphy of NE Poland or/and SW edge of ancient Baltica. On a regional scale, their emplacement coincides with a lithospheric extension and with a prominent pulse of thermal activity, related to the Late Variscan time (at ~300 Ma), heating of the Ediacaran and Carboniferous sedimentary succession before the Permian.

**Supplementary Materials:** The following are available online at https://www.mdpi.com/article/10.3390/min11121361/s1, Table S1: U-Pb zircon SHRIMP analytical data from Mielnik IG1 dolerite sample; Table S2: EPMA composition of selected mineral phases (A) clinopyroxene, (B) Feldspars, (C) Magnetite, (D) Titanite; Table S3: Major and trace element contents in dolerite from the Mielnik IG1 borehole.

**Author Contributions:** Conceptualization, P.P., J.P. and E.K.; methodology, E.K. and L.K.; investigation, E.K., L.K., P.P., J.P. and K.N.; data curation, L.K. and E.K.; writing—original draft preparation, E.K. and P.P.; writing E.K.; visualization, L.K., P.P., J.P. All authors have read and agreed to the published version of the manuscript.

**Funding:** This study and final editorial work were financially supported by Polish Geological Institute–National Research Institute (PGI–NRI) internal grant number 62.9012.2165.00.0; Publication fee was financed by The National Fund for Environmental Protection and Water Management.

**Data Availability Statement:** The data presented in this study are openly available in Supplementary Material.

**Acknowledgments:** Thanks are addressed to Dominik Gurba (PGI-NRI) for zircon separation, mount preparation and CL imaging and to Zbigniew J. Czupyt for SHRIMP IIe/MC calibration procedure and support. Constructive comments by two anonymous reviewers greatly improved the manuscript.

**Conflicts of Interest:** The authors declare no conflict of interest.

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
