# Peer review of "First Evidence of the Post-Variscan Magmatic Pulse on the Western Edge of East European Craton: U-Pb Geochronology and Geochemistry of the Dolerite in the Lublin Podlasie Basin, Eastern Poland"

_minerals, doi:10.3390/min11121361_

Round 1
Reviewer 1 Report
please see attached

Author Response
Dear Reviewer #1
We are grateful for the effort that you dedicated to providing all the remarks on our manuscript.
We are appreciative for the insightful comments and valuable improvements to our paper.
We have addressed all the suggestions
The corrected version of manuscript includes:
- a new version of Figure 6: two diagrams (6G and 6H) presented a trace elements contents in zircons have been added as it was suggested
- a new version of Figure 10, where on the spider diagram (10B) the composition of average Upper Continental Crust has been presented as a reference to show directly a scope of crustal contamination detected in Mielnik IG1 dolerite;
Below we state how we address each comment in a revised manuscript. Original comments are in black; our responses are indented and in blue font, for a point-by-point response to the comments and concerns.
All numbers of lines refer to the version manuscript, that has been reviewed. There are two files in the attachment “clean” and file with tracked changes.
Reviewer 1
This is a document of regional interest. It is acceptably written and the data presented are sound,
although some slight differences in interpretation are possible. Accept with very minor changes.
(1) Zircons in basalts. Alarm Bells ring. Zircons should not be there, unless there is sufficient
crystallization to produce relatively siliceous magma pockets. The authors do try to address this
worry. The zircons presented look like a detrital suite. Wonder if they are ALL inherited. Then the
only change of logic would be Carboniferous-Permian granites are at depth (to be inherited), and that the zircon age is a maximum emplacement age.
(2) It would be fun to look at the geochemistry of the zircons using ICPMS to see if the young grains carry the earmarks of mafic or felsic compositions. Yes the U contents of the zircons are moderately low, lower than that of an average granite.
(3) One of the important parts of the paper is the record of regional heating/reheating. List this in the
conclusions.
(4) Dikes and sill represent very different stress regimes. There should be a comment on this in the
manuscript.
(5)The titanite should be a U-Pb datable phase even though they are probably post-magmatic.
(6) Anywhere Fe-oxides are referred to should be Fe-Ti-oxides.
(7) Some information on the original drill hole would be nice.
(1) We agree with the reviewer it is very difficult to date a basalt, but there are notable exceptions to this. One of the example is, the work on the basalt flows of King George Island by Nawrocki and others 2010. (2010) - Isotopic ages and palaeomagnetism of selected magmatic rocks from King George Island (Antarctic Peninsula). J. Geol. Soc. London, 167: 1063-1079 Mentioned contribution provided a results of U-Pb dating of zircon from volcanic rocks (including also basalts composition according to TAS classification). These results are supported by Ar-Ar data.
Nerveless the discussion part related to geochronology has been extended.
Dolerite from the Mielnik (of sub-alkaline basalt composition according to Winchester& Floyd classification) is a sub-volcanic / rock with longer time of crystallization. Moreover the geochemistry indicate that it is not a primordial melt, but enriched and contaminated. Thus limited zircons presumably were formed at a late stage of igneous crystallization of dolerite in Zr-enriched interstitial melt pockets.
(2) According to the suggestion some of trace elements in zircons (including youngest group) have been plotted on two discrimination diagrams. They are presented as a Figure 6 G, H. These trace elements concentrations expose a very similar trend of the youngest group, as has been defined by composition of zircons hosted in A type magmatic rock (granitoids) in contrast to plots of the older xenocristic grains, that better correspond to the field of S –type magmatism diagrams after Wang et al. 2012.
(3) – Thank you for pointing this out – is added to conclusion.
(4)- the nomenclature has been unified in the manuscript Mielnik dolerite is a dyke (e.g. dykes are often associated with extensional settings); The “dykes” are sometimes referred to as sille and that original interpretation have been preserved, e.g. the Neoproterozoic CFB _Shumlyanskyy et al. 2016 (sills are known in rift-related sedimentary basins; they are important in the breakup of continents and the production of flood basalts; or Great Sill from Great Whin Dolerite Complex (Hamilton et al. 2011).
(5) Indeed; a single grains of the titanite mount, originally was prepared for U-Pb SHRIMP age measurements. The epoxy mount also contains the reference material - grains of the Khan reference standard titanite ( U= from 500 to 819 ppm). The Mielnik IG1 titanite revealed however a relatively low uranium concentration ( ~60 -120 ppm). In such case a low U content always results in high measurement errors and then poorly U-Pb result.
(6) It has been corrected;
(7) Some more details has been added.
Specific comments
Line -33…complicated by faults, particularly rift zones.
Ekrz: This sentence has been completed
Line-33-35 Their distribution provides information about the spatial and temporal changes of ancient crustal structures, the regional stress field, their intensity and reasons for them.
Ekrz: this sentence has been corrected (completed)
Line 47…Ediacaran-early Palaeozoic, Devonian-Carboniferous, and Permian-Mesozoic sediments, [is this correct]
Ekrz: This has been corrected (replaced)
Line 63-65: This contribution presents the only U-Pb zircon ages and whole-rock geochemical characteristics for the dolerite previously known from the Mielnik IG1 drill hole, in North-Eastern Poland in the vicinity of the Teisseyre-Tornquist margin.
Ekrz: It has been corrected (removed)
Line 66: U-Pb SHRIMP zircon measurements, the ~18m thick dolerite sill in the Proterozoic
Ekrz: It has been corrected (replaced)
Line 78-79 hosted several episodes of continental rift-related magmatism, resulting in the emplacement of dolerite dykes.
Ekrz: it has been corrected
Line 80-82…is the Central Scandinavian Dolerite group. These mafic dykes intruded basement rock between 1271–1246 Ma probably in response to the distal events of Mesoproterozoic subduction along the westernmost margin
Ekrz: It has been corrected
Line 90-95 Doleritic sills located immediately beneath the Volyn flood basalt sequence, were identified in Belarus and Ukraine [24-25, 6], have limited radiometric dating. An olivine dolerite yielded a U-Pb baddeleyite age of 626 ± 17 Ma by (XXXtehcnique) [6], which is an age that predates the main volcanic phase of the Volyn Flood Basalt Province.
Ekrz: It has been corrected
So you are working on dikes and these are sills. How do you relate the change of geometry and
crustal stress?
EKrz: The dolerite recovered by drilling in Mielnik IG1 is a dyke. The Late Neoproterozoic dolerite sills were described by Shumlyanskyy et al, 2016 and their interpretation were taken from the cited work as literally as possible ( “High-Ti dolerite sills geochemically close to the high-Ti tholeiite basalts are rather common immediately beneath the Volyn flood basalt sequence.” of Volyno-Podolian monocline, Ukraine - (Shumlyanskyy et al., 2016, GFF 138)
Line 101-106
The magma injections piercing the Paleozoic sedimentary cover are not commonly exposed in this part of the EEC margin, but a few single dolerites and sills have been identified [27]. Some of them are located near the TTZ in NE Poland (Figure 2), in the vicinity of the prominent Carboniferous intra-cratonic alkaline and ultramafic-alkaline massifs of e.g. Ełk, Pisz, Tajno, [28–30]. They appear as several meter thick intrusions between the Silurian and Rotliegend sediments.
Ekrz: Even the Palaeozoic ignieous rocks in this part of EEC are accessible only by deep drillings, thus this is difficult to use “exposed”. “ Nevertheless the sentence has been simplified
Vein, I believe, is a term restricted to hydrothermal deposits. Stick with dikes and sills.
Ekrz Thank you for pointing this out. It has been corrected.
Line 115-116
Further south isolated volcanic rocks (alkaline basalts) have been identified in the lowermost part of Carboniferous succession.
Ekrz: It has been corrected
- Geological context of the Mielnik IG1 dolerite occurrence Who drilled it and when?
Ekrz: these drilling details have been added.
Line 133-135
In this part of the EEC the dominant evidence of igneous activity remains in the presence of the volcanogenic Sławatycze Formation, in E Poland, and the analogous Volyn Seriess in Ukraine [6–7].
Ekrz: The name Volyn Series is correct
Line 142-143
The initial rift-related sediments (Figure 3) are alluvial basal conglomerates and coarse-grained
sandstones, referred to as the Żuków Formation,
Ekrz: This sentence has been corrected
Line 146-147
There is a large volume of basaltic lava flows 10 to 30 meters thick.
Ekrz: this sentence has been corrected
Line 150
Towards the top of the succession, effusive eruptions are less common.
Ekrz: This sentence has been corrected
Line 151-152
The seven identified explosive episodes represent a large volume of proximal pyroclastic material
(XXX km3).
Ekrz: this sentence has been corrected. A “volume” is not added. The volume of pyroclstics has not been estimated (and still remains unknown). The Ediacaran volcanogenic deposits remain hidden. They are recognized only by several deep drilling. A volume of the pyroclastic sediments was related also to the local rift tectonics of the Lublin Podlasie basin, so it is impossible to estimate a total their volume.
Line 157-158
Łopiennik and Włodawa formations [47–47]. Check refs
Ekrz: Thank you for pointing this out. It has been checked and corrected.
Line 162-163
During the Ordovician, deposition continued in form of relatively thin shallow marine clastic and
carbonate sediments overlain by Silu-
Ekrz: This sentence has been corrected
Lines 169-176
In the Mielnik IG1 section, a hypabyssal mafic rock lies between the top of the Paleoproterozoic
(1.86 Ga) crystalline basement rocks, at a depth of 1746 m, but above the clearly identified Żuków
Formation (depth 1728 m). The dolerite lies above as well as at the bottom of Volyn volcanic series,
at the depth of 1722 m (Figure 3). Initial descriptions assigned tentative ages to the drill core
formations. A crucial contact between the Mielnik IG1 dolerite and the Żuków Formation
conglomerate was not properly identified/described nor maintained (Figure 4A) leading to confusion
in the literature about its likely age [19–20, 48–49].
Ekrz: This paragraph has been corrected according to Reviewer suggestion.
Line 178
with randomly oriented plagioclase laths enclosed by clinopyroxene (augite) and Fe
Ekrz: This sentence has been corrected
Line 179-180
An archival description [26] indicates finer-grained forms at the bottom contact of sill (interpreted as a chilled margin). [is this true? Finer near the contact???]
Ekrz- this sentence has been omitted – because it is quoting an opinion referred in [26]
Line 184
Fig 4 C polished, really? Looks very rough.
Ekrz: The Mielnik IG1 dolerite is an altered rock (with LOI between 4.20 - 6.00 wt%), partially porous, due to the different degree of preservation of mineral components (olivine are completely altered, plagioclase are albitized).
In this context of altered rock- forming components it is difficult to get a perfect polish. It is undoubtedly not a natural surface, but after a polishing process.
Line 202
primary O2–ion beam at 3.5 nA, with 20-23 μm spot size . For each analysis, six peak
Ekrz: corrected
Line 252
These data are mostly discordant, but there is one grain (14.1) with a relatively low level
Ekrz: corrected
Line 321
uncommonly observed in magmatic titanites e.g lamprophyres [61]. Morad et al. [66] sug-
Ekrz: corrected
Line 334-335
The incorporation of Zr into titanite has been found to be strongly dependent on temperature [69]. The measured maximum contents indicate the crystallization temperature
Ekrz: corrected
533-534
value of 143Nd/144Nd of the Mielnik IG1 sample undoubtedly represents enriched mantle sources of EM I type.
Some would say it is undoubtedly crustally contaminated and that is all.
Ekrz We agree with the Reviewer opinion and this paragraph has been corrected.
Ewa Krzemińska (Ekrz)
as a corresponding author

Reviewer 2 Report
Dear Authors,
Paper authored by Ewa Krzemińska and colleagues focused on the Paleozoic magmatism of the SW margin of East European craton (EEC). Authors present new age, petrography, mineralogy, geochemical data for dolerite sill from the Mielnik borehole. Novelty of the article is related with dating the Mielnik sill in at 300 ± 4 Ma. The revised age of dolerite indicate the widespread, including the margin of the EEC, occurrence of the Permo-Carboniferous magmatic activity. The topic of article will be of interest to the readers who studying tectonics of EEC.
The composition of the article is logical. References are relevant.
Despite the positive view on the paper, I have several comments aimed at improving the manuscript. The main problem is that some of the authors ‘conclusions do not distinctively support by the data. This is mainly because the Mielnik sill rocks are very strongly altered and contaminated, which masks the original characteristics of the melt. Keeping that in mind, the authors should have been more careful in their conclusions.
I think that the article can be published after correction by the authors according to the comments below
Major Comments
1.The Methods description excluding zircon dating is scarce. Please, give the microprobe and XRF technique in detail.
2.The reviewer has no doubt that the zircon age yielded by the authors at 300 Ma corresponds to the magmatic episode in which the emplacement of the Mielnik dolerite occurred. At the same time, I cannot agree with the author's without- alternative interpretation of zircon as crystallized from dolerite melt.
It is hardly to agree that a zircon has cristallized from silica- undersaturated (45.50.86–49.59 wt% SiO2) and low-Zr (110 – 127 ppm) melt as the Mielnik dolerite. Zircon, originated from basaltic melt has an elongated prismatic shape, but the zircon from Mielnik dolerite is not. The authors say that the zircon has precipitate from late evolved melt (line 451). If it was the case, the zircon must have had an enriched signature. However, the zircon has very low U concentration that is clearly shown by CL images in Fig. 5E.
I think that at least two alternative hypotheses of the zircon origin can be discussed. First, the zircons could have been trapped by the dolerite melt from a magma chamber with a coeval more silica-rich melt. For example, that melt could have been formed by melting of crustal rocks under the heat of basaltic melts.
The second hypothesis, (more realistic) is that some strongly metamictic grains of inherited zircon were annealing when they were hosted by the dolerite melt. As a result, zircon U-Pb isotopic system was re-equilibrated on an age of this event. A broadly similar case is described in Bogdanova et al., 2021.
I strongly recommend discussing the origin of young zircons in the Mielnik dolerite, not exclusively they are autocrysts but also other possible points of view.
Minor Comments
Line 40 1.2 Ga the Lentiira–Kuhmo–Kostomuksha lamproites (Finland and Karelia, Russia) (e.g. Kargin et al., 2014, Dalton et al., 2019) would be nice to complete the Mesoproterozoic magmatic episode.
Line 84-85 In addition to Egersund there are some Neoproterozoic dyke swarms hosted in the Caledonian Nappes associated with opening of the Japetus ocean (e.g. Tegner et al., 2019; Kumpulainen et al., 2021 and references therein)
Line 102. What is single dolerite? Please, give explanation.
Line 197– please, let the abbreviation “PIG” be deciphered once (Polish Geological Institute) in the Methods section or give the abbreviation “PIG-NGI” in affiliation 1
Line 106 Please, correct the phrase “The dolerite vein emplacement ages and of other subvolcanic rocks” I think it should be “The dolerite vein emplacement ages and ones of other subvolcanic rocks”
Line 276-277. Please, leave the apatite in one position only – prominent or other accessory mineral
Line 301. Replace “titanomaghemite field” by “titanohematite field”
Line 313. add a space to “forming irregular-shaped grains”
Line 327-328. You show Y concentration (90 ppm) that is low then a detection limit
Lines 422, 431 – “autocrysts” – see Major Comments
Lines 451-458 I think to calculate a zircon saturation temperature for strongly altered dolerite is not a great idea. This approach applies to the range of felsic to intermediate melts – see Boehnke et al., 2013 and references therein.
Lines 532-534. Please, remove “The low value of 143Nd/144Nd of the Mielnik IG1 sample undoubtedly represents enriched mantle sources of EM I type”. I recommend to leave a suggestion that the low value of 143Nd/144Nd of the Mielnik IG1 sample represent “the influence of contamination of its parent melt by the material of the continental crust”. It is undoubtedly.
Lines 538-541. There is not enough argument for this conclusion about a mantle source. I recommend a more conservative conclusion, keeping in mind that the Mielnik dolerite is highly altered and contaminated rock.
References
Hayden Dalton, Andrea Giuliani, Hugh O’Brien, David Phillips, Janet Hergt, Roland Maas, Petrogenesis of a Hybrid Cluster of Evolved Kimberlites and Ultramafic Lamprophyres in the Kuusamo Area, Finland, Journal of Petrology, Volume 60, Issue 10, October 2019, Pages 2025–2050, https://doi.org/10.1093/petrology/egz062
Tegner, C., Andersen, T.B., Kjøll, H.J., Brown, E.L., Hagen-Peter, G., Corfu, F., Planke, S., Torsvik, T.H., 2019. A Mantle Plume Origin for the Scandinavian Dyke Complex: A “Piercing Point” for 615 Ma Plate Reconstruction of Baltica? Geochem. Geophys. Geosyst. 20, 1075–1094. https://doi.org/10.1029/2018GC007941
Kumpulainen, R.A., Hamilton, M.A., Söderlund, U. & Nystuen, J.P., U-Pb baddeleyite age for the Ottfjället Dyke Swarm, central Scandinavian Caledonides: new constraints on Ediacaran opening of the Iapetus Ocean and glaciations on Baltica. GFF, Vol. XXX, pp. XX-XX. © Geologiska Föreningen. http://dx.doi.org/10.1080/11035897.
Svetlana V. Bogdanova, Elena Belousova, Bert De Waele, Alexander N. Larionov, Sandra Piazolo, Alexander V. Postnikov, Alexander V. Samsonov Palaeoproterozoic reworking of early Archaean lithospheric blocks: Rocks and zircon records from charnockitoids in Volgo-Uralia // Precambrian Research 360 (2021) 106224 https://doi.org/10.1016/j.precamres.2021.10622420
Patrick Boehnke, E. Bruce Watson, Dustin Trail, T. Mark Harrison, Axel K. Schmitt, Zircon saturation re-revisited, Chemical Geology, Volume 351, 2013, Pages 324-334, ISSN 0009-2541, https://doi.org/10.1016/j.chemgeo.2013.05.028
Author Response
Dear Reviewer# 2
We are grateful for the dedicated to providing and all remarks on our manuscript. We are appreciative for the insightful comments that allow to improve to our paper.
We have incorporated almost all of the suggestions
The corrected version of manuscript includes:
- a new version of Figure 6: two diagrams (6G and 6H) presented a trace elements contents in zircons have been added to support our conclusions.
- a new version of Figure 10, where on the spider diagram (10B) the composition of average Upper Continental Crust has been presented as a reference to show directly a scope of crustal contamination detected in Mielnik IG1 dolerite;
Below we state how we address each comment in a revised manuscript. Original comments are in black; our responses are indented and in blue font, for a point-by-point response to the comments and concerns.
All numbers of lines refer to the version manuscript, that has been reviewed. There are two files in the attachment “clean” and file with tracked changes.
Review Report (Reviewer 2)
Comments and Suggestions for Authors
Dear Authors,
Paper authored by Ewa Krzemińska and colleagues focused on the Paleozoic magmatism of the SW margin of East European craton (EEC). Authors present new age, petrography, mineralogy, geochemical data for dolerite sill from the Mielnik borehole. Novelty of the article is related with dating the Mielnik sill in at 300 ± 4 Ma. The revised age of dolerite indicate the widespread, including the margin of the EEC, occurrence of the Permo-Carboniferous magmatic activity. The topic of article will be of interest to the readers who studying tectonics of EEC.
The composition of the article is logical. References are relevant.
Despite the positive view on the paper, I have several comments aimed at improving the manuscript. The main problem is that some of the authors ‘conclusions do not distinctively support by the data. This is mainly because the Mielnik sill rocks are very strongly altered and contaminated, which masks the original characteristics of the melt. Keeping that in mind, the authors should have been more careful in their conclusions.
I think that the article can be published after correction by the authors according to the comments below.
Major Comments
- The Methods description excluding zircon dating is scarce. Please, give the microprobe and XRF technique in detail.
Ekrz: Thank you for pointing out this.This section has been corrected and extended. Besides, we found out mistake. After checking the description from the laboratory, we have to change the method details: –the major elements were analyzed by ICP-OES.
- The reviewer has no doubt that the zircon age yielded by the authors at 300 Ma corresponds to the magmatic episode in which the emplacement of the Mielnik dolerite occurred. At the same time, I cannot agree with the author's without- alternative interpretation of zircon as crystallized from dolerite melt.
Ekrz: The part of the discussion related to the age investigation has been changed. It has a different title ( 6.1 Age interpretation). The zircons trace element contents and the features of youngest zircon cluster have been better exposed and discussed. The impact of the strong contamination has been underlined, also in geochemical chapter by adding to the spider diagram (Figure 10B) a pattern of an average Upper Continental Crust to show directly a an evidence of the crustal contamination.
- 3. It is hardly to agree that a zircon has cristallized from silica-undersaturated (45.50.86–49.59 wt% SiO2) and low-Zr (110 – 127 ppm) melt as the Mielnik dolerite. Zircon, originated from basaltic melt has an elongated prismatic shape, but the zircon from Mielnik dolerite is not. The authors say that the zircon has precipitate from late evolved melt (line 451). If it was the case, the zircon must have had an enriched signature. However, the zircon has very low U concentration that is clearly shown by CL images in Fig. 5E.
Ekrz: Indeed the mafic composition of Mielnik dolerite should limit zircon saturation. We improved the arguments and expanded the discussion. We found inspiration in the publication of Davies et al. 2021, Their concluded:
“Zircon crystallizes in late-stage enriched melt pockets in LIP magmas. It is likely that crustal contamination of the melts is required for zircon to crystallize in many samples of the CAMP and also for LIP magmas in general. Contamination by upper crustal sediments such as shales most easily creates the conditions for crystalizing zircon…”
.Davies et al. Zircon petrochronology in large igneous provinces reveals uppercrustal contamination processes: new U–Pb ages, Hf and O isotopes, and trace elements from the Central Atlantic magmatic province (CAMP)Contributions to Mineralogy and Petrology (2021) 176:9
https://doi.org/10.1007/s00410-020-01765-2
The uranium contents of 98ppm and then 142 – 550 ppm in youngest zircons of Mielnik dolerite support our conclusion. Moreover the morphological forms of zircon are more fitted to interstitial origin.
I think that at least two alternative hypotheses of the zircon origin can be discussed. First, the zircons could have been trapped by the dolerite melt from a magma chamber with a coeval more silica-rich melt. For example, that melt could have been formed by melting of crustal rocks under the heat of basaltic melts. The second hypothesis, (more realistic) is that some strongly metamictic grains of inherited zircon were annealing when they were hosted by the dolerite melt. As a result, zircon U-Pb isotopic system was re-equilibrated on an age of this event. A broadly similar case is described in Bogdanova et al., 2021.
I strongly recommend discussing the origin of young zircons in the Mielnik dolerite, not exclusively they are autocrysts but also other possible points of view.
Ekrz: The firs crucial part of discussion related to age data interpretation has been extended (as 6.1 Age interpretation). To recognize the origin zircons -their trace elements contents (including youngest group) have been plotted on two discrimination diagrams. They are presented as a Figure 6 G, H. These trace elements concentrations expose a very similar trend of the youngest group, as has been defined by composition of zircons hosted in A type magmatic rock (granitoids) in contrast to plots of the older xenocristic grains, that better correspond to the field of S –type magmatism diagrams after Wang et al. 2012.
Taking into account the coherent composition of youngest zircons with moderate uranium content < 500ppmwe did not consider their hypothetical prior metamictization.
Minor Comments
Line 40 1.2 Ga the Lentiira–Kuhmo–Kostomuksha lamproites (Finland and Karelia, Russia) (e.g. Kargin et al., 2014, Dalton et al., 2019) would be nice to complete the Mesoproterozoic magmatic episode.
Ekrz: Indeed, however, these are quite specific types of magmatism associated with kimberlites. Moreover, in this part of the text, the presented examples have been narrowed to the SW margin of the Baltica.
Line 84-85. In addition to Egersund there are some Neoproterozoic dyke swarms hosted in the Caledonian Nappes associated with opening of the Japetus ocean (e.g. Tegner et al., 2019; Kumpulainen et al., 2021 and references therein);
Ekrz This part of the text has been supplemented with the publication of Kumpulainen et al., 2021, that provides a new age data from the The Ottfjället Dyke Swarm.
Line 102. What is single dolerite? Please, give explanation.
Ekrz: the sentence has been corrected and the explanation from referred contribution has been added.
Line 197– please, let the abbreviation “PIG” be deciphered once (Polish Geological Institute) in the Methods section or give the abbreviation “PIG-NGI” in affiliation 1
Ekrz: it has been completed.
Line 106 Please, correct the phrase “The dolerite vein emplacement ages and of other subvolcanic rocks” I think it should be “The dolerite vein emplacement ages and ones of other subvolcanic rocks”
Ekrz: it has been corrected.
Line 276-277. Please, leave the apatite in one position only – prominent or other accessory mineral
Ekrz: it has been corrected.
Line 301. Replace “titanomaghemite field” by “titanohematite field”
Ekrz: it has been corrected.
Line 313. add a space to “forming irregular-shaped grains”
Ekrz: it has been corrected.
Line 327-328. You show Y concentration (90 ppm) that is low then a detection limit
Ekrz: Thank you for pointing this out. Indeed the detection limit is 104 ppm –this part of data presentation has been corrected.
Lines 422, 431 – “autocrysts” – see Major Comments
Ekrz the wording “autocryst “ has been removed
Lines 451-458 I think to calculate a zircon saturation temperature for strongly altered dolerite is not a great idea. This approach applies to the range of felsic to intermediate melts – see Boehnke et al., 2013 and references therein.
Ekrz: Indeed. This part of paragraph has been removed.
Lines 532-534. Please, remove “The low value of 143Nd/144Nd of the Mielnik IG1 sample undoubtedly represents enriched mantle sources of EM I type”. I recommend to leave a suggestion that the low value of 143Nd/144Nd of the Mielnik IG1 sample represent “the influence of contamination of its parent melt by the material of the continental crust”. It is undoubtedly.
Ekrz: it has been corrected.
Lines 538-541. There is not enough argument for this conclusion about a mantle source. I recommend a more conservative conclusion, keeping in mind that the Mielnik dolerite is highly altered and contaminated rock.
Ekrz_ We agree with the Reviewer. The available geochemical signature is completely blurred by contamination. this part of the conclusions has been modified.
Ewa Krzemińska
As a corresponding author

Round 2
Reviewer 2 Report
Dear Authors,
I think that after the corrections you have made, the paper can be recommended for publication